# Decoupled Offline to Online Finetuning via Dynamics Model

## Abstract

Constrained by the sub-optimal dataset in offline reinforcement learning (RL), the offline trained agent should be online finetuned before deployment. Due to the conservative offline algorithms and unbalanced state distribution in offline dataset, offline to online finetuning faces severe distribution shift. This shift will disturb the policy improvement during online interaction, even a performance drop. A natural yet unexplored idea is whether policy improvement can be decoupled from distribution shift. In this work, we propose a decoupled offline to online finetuning framework using the dynamics model from model-based methods. During online interaction, only dynamics model is finetuned to overcome the distribution shift. Then the policy is finetuned in offline manner with finetuned dynamics and without further interaction. As a result, online stage only needs to deal with a simpler supervised dynamics learning, rather than the complex policy improvement with the interference from distribution shift. When finetuning the policy, we adopt the offline approach, which ensures the conservatism of the algorithm and fundamentally avoids the sudden performance crashes. We conduct extensive evaluation on the classical datasets of offline RL, demonstrating the effective elimination of distribution shift, stable and superior policy finetuning performance, and exceptional interaction efficiency within our decouple offline to online finetuning framework.

## 1 Introduction

As an approach closely aligned with data-driven paradigms, offline reinforcement learning (Levine et al., 2020) has ignited the enthusiasm of the community. A large number of algorithms have been developed with remarkable speed, encompassing not only traditional RL algorithms that are designed to overcome overestimation on out-of-distribution (OOD) state-action pairs (Kumar et al., 2020; Fujimoto & Gu, 2021; Zhuang et al., 2023), but also supervised paradigms such as sequence modeling (Chen et al., 2021; Zhuang et al., 2024). Due to the limitations of data quality, the policies obtained from offline learning may not be optimal and are challenging to directly deploy in real-world scenarios. This has given rise to the problem of offline to online finetuning (Guo et al., 2023; Nakamoto et al., 2024), which aims to further improve the performance through online interaction.

Offline to online finetuning faces the challenge of distribution shift, which is caused by the unbalanced state distribution in the offline dataset (Fu et al., 2020) and the inherent conservatism of offline algorithms (Kumar et al., 2020). Online finetuning pursues the superior performance than offline pretrained policy and the exploration on out-of-distribution region that may yield high return is unavoidable. Such exploration may be blind or even dangerous, such as sudden performance drops (Nakamoto et al., 2024; Lyu et al., 2022), unless the distribution shift has been eliminated. That is, offline to online finetuning is required to address two conflicting issues: policy optimization and distribution shift elimination. Existing methods can be broadly categorized into two classes: Some algorithms (Nakamoto et al., 2024; Wu et al., 2022; Lyu et al., 2022), less conservative in nature, have been crafted in an attempt to mitigate the extent of distribution shift, yet they are powerless against the inherently unbalanced distribution within the offline dataset. Other algorithms address the distribution shift by imposing additional constraints (Lee et al., 2022; Li et al., 2023), making the policy safer and more effective when exploring OOD regions. Regardless, the elimination of distribution shift and policy improvement are perpetually intertwined, with a compromise and trade-off that must exist. A natural question thus arises:

*Can we decouple the elimination of distribution shift from policy improvement?*

If so, we might first eliminate the distribution shift and then carry on the policy improvement. Such an approach could fundamentally avoid conflicts and the associated compromises, maximizing the capabilities of both distribution shift elimination algorithms and policy improvement.

Within the context of model-based offline algorithms (Janner et al., 2019; Yu et al., 2020), we propose a framework named **D**ecoupled **O**ffline to **O**nline **F**inetuning (DOOF). DOOF has successfully decouples the elimination of distribution shift from policy improvement. Specifically, during the online interaction phase, we focus solely on the elimination of distribution shift by finetuning a more accurate dynamics model. Subsequently, we utilize this finetuned dynamics model to assist in the finetuning of the policy in an offline mode, without the need for further interaction with the environment. In this way, the online phase only needs to address a simpler supervised dynamics model learning, rather than more complex policy improvement affected by the distribution shift. In addition, we leverage the model uncertainty from MOPO to encourage the data collection on OOD regions where the distribution difference is significant and the dynamics prediction is inaccurate. The policy is finetuned through offline algorithms, which indicates that the inherent conservatism is retained and the algorithmic consistency is ensured, fundamentally avoiding sudden policy collapse. We validate our algorithm on the classic datasets from D4RL (Fu et al., 2020) and find that with only 10k online interaction steps and 300k offline training steps, DOOF achieves significant performance improvements. In contrast, other baseline models fail to improve but have also experienced a decline in performance in some cases. Such exceptionally high interaction efficiency is attributed to the simplification of the online phase and efficient exploration guided by the dynamics uncertainty.

## 2 PRELIMINARY

### 2.1 OFFLINE REINFORCEMENT LEARNING

Reinforcement learning (RL) is typically formulated by a Markov Decision Process (MDP) $\mathcal{M} = \{\mathcal{S}, \mathcal{A}, r, P_{\mathcal{M}}, d_0, \gamma\}$, with state space $\mathcal{S}$, action space $\mathcal{A}$, scalar reward function $r(s_t, a_t)$, transition dynamics $P_{\mathcal{M}}(s_{t+1}|s_t, a_t)$, initial state distribution $d_0(s_0)$ and discount factor $\gamma$ (Sutton et al., 1998). The objective of RL is to optimize a policy $\pi(a_t|s_t)$ that maximize the expectation of discounted return $J(\pi, \mathcal{M}) = \mathbb{E}_{\tau \sim P_{\pi, \mathcal{M}}(\tau)}\left[\sum_{t=0}^{T} \gamma^t r(s_t, a_t)\right]$, where $P_{\pi, \mathcal{M}}(\tau) = d_0(s_0) \prod_{t=0}^{T} \pi(a_t|s_t) P_{\mathcal{M}}(s_{t+1}|s_t, a_t)$ is the distribution of trajectory $\tau$ generated from the interaction between the policy $\pi(a_t|s_t)$ and the environment $\mathcal{M}$. The value function $V_{\mathcal{M}}^{\pi}(s) = \mathbb{E}_{\tau \sim P_{\pi, \mathcal{M}}(\tau|s, a)}\left[\sum_{t=0}^{T} \gamma^t r(s_t, a_t)|s_0 = s\right]$ gives the expected discounted return under policy $\pi$ when starting from state $s$ in environment $\mathcal{M}$.

**Offline** reinforcement learning forbids (Levine et al., 2020) the interaction with the environment $\mathcal{M}$ and only a fixed offline dataset $\mathcal{D} = \{(s_t, a_t, r_t, s_{t+1})\}_{t=1}^{N}$ is provided. This setting is more challenging since the agent is unable to explore the environment and collect additional feedback. This will lead to overestimation on out-of-distribution state-action pairs, resulting in terrible performance.

### 2.2 MODEL-BASED OFFLINE RL ALGORITHMS

Existing model-based offline RL methods are designed based on model-based policy optimization (MBPO) (Yu et al., 2020). MBPO can be divided into two stages, transition dynamics pretraining and policy learning. During transition dynamics pretraining, MBPO estimates an dynamics model[1] $P_{\widehat{\mathcal{M}}}$ from the online replay buffer or the offline dataset $\mathcal{D}$ using maximum likelihood estimation:

$$P_{\widehat{\mathcal{M}}} = \arg\min_{P_{\widehat{\mathcal{M}}}} \mathbb{E}_{(s_t, a_t, s_{t+1}) \sim \mathcal{D}}\left[-\log P_{\widehat{\mathcal{M}}}(s_{t+1}|s_t, a_t)\right]. \tag{1}$$

Usually, the dynamics model is considered as a neural network that predicts a Gaussian distribution $P_{\widehat{\mathcal{M}}}(s_{t+1}|s_t, a_t) = \mathcal{N}(\mu_\theta(s_t, a_t), \Sigma_\phi(s_t, a_t))$. Besides, this dynamics model is actually an en-

---

[1]Here we assume the reward function $r$ is known. If not, the reward can be considered as part of the dynamics model $P_{\widehat{\mathcal{M}}, r}(s_{t+1}, r_t|s_t, a_t)$. Besides, the following theoretical analysis can also applied to the situation with unknown reward function (Yu et al., 2020).

semble model when implementation $\{P_{\widehat{\mathcal{M}}}^k = \mathcal{N}(\mu_\theta^k, \Sigma_\phi^k)\}_{k=1}^K$. With the learned dynamics model $P_{\widehat{\mathcal{M}}}$, we can construct an estimated MDP $\widehat{\mathcal{M}} = \{\mathcal{S}, \mathcal{A}, r, P_{\widehat{\mathcal{M}}}, d_0, \gamma\}$.

Thereafter, MBPO utilizes a standard actor-critic RL algorithm SAC (Haarnoja et al., 2018) to recover optimal policy with the help of the estimated MDP $\widehat{\mathcal{M}}$. An augmented dataset $\mathcal{D} \cup \mathcal{D}_{\widehat{\mathcal{M}}}$ is used to train the policy, where $\mathcal{D}_{\widehat{\mathcal{M}}}$ is synthetic data generated by performing $h$-step rollouts in $\widehat{\mathcal{M}}$ starting from states in $\mathcal{D}$. During policy training, mini-batches are drawn from $\mathcal{D} \cup \mathcal{D}_{\widehat{\mathcal{M}}}$, where each datapoint is sampled from the real data $\mathcal{D}$ with the probability $p$, and from $\mathcal{D}_{\widehat{\mathcal{M}}}$ with probability $1 - p$. Model-based offline policy optimization (MOPO) Yu et al. (2020) proposes to penalize the reward function by the uncertainty $u(s, a)$ of the learned dynamics models:

$$\hat{r}_t(s, a) = r_t(s, a) - \lambda u(s, a), \tag{2}$$

where penalty coefficient $\lambda$ is a hyperparameter and the uncertainty $u(s, a)$ is usually empirically and lacks theoretical guarantee (Yu et al., 2020; Lu et al., 2021). While MOBILE (Sun et al., 2023) theoretically conducts uncertainty quantification through the inconsistency of Bellman estimations.

### 2.3 OFFLINE TO ONLINE FINETUNING

In this paper, we only consider offline RL with datasets that comprise sub-optimal trajectories rather than optimal ones. If optimal, naive supervised methods such as behavior cloning (Pomerleau, 1988) would sufficient to learn an optimal policy, which is not the issue that offline RL aims to address.

**Definition 2.1 (Offline to Online Finetuning)** *Assume the offline dataset $\mathcal{D}$ is sub-optimal and the offline pretrained policy $\pi_{off}$ still sub-optimal.* **Offline to Online Finetuning** *aims to further improve the performance of $\pi_{off}$ through the interaction with environment $\mathcal{M}$.*

**Challenges** The entire state-action space $(s, a)$ can be divided into three distinct segments based on the alignment of state and action distributions with those from the offline dataset: a) in-distribution (ID) state-action pairs $(s_{\mathcal{D}}, a_{\mathcal{D}})$ with $s \sim \mathcal{D}, a \sim \mathcal{D}$, b) in-distribution (ID) state but out-of-distribution (OOD) action pairs $(s_{\mathcal{D}}, a_{\neg\mathcal{D}})$ with $s \sim \mathcal{D}, a \not\sim \mathcal{D}$, c) totally out-of-distribution (OOD) state-action pairs $(s_{\neg\mathcal{D}}, a)$ with $s \not\sim \mathcal{D}, a \sim \mathcal{A}$.

To guarantee the performance of $\pi_{off}$, offline algorithms enforce the conservative Q-value on ID state but OOD action pairs $(s_{\mathcal{D}}, a_{\neg\mathcal{D}})$ to prevent choosing the OOD action $a_{\neg\mathcal{D}}$ given ID state $s_{\mathcal{D}}$ (Zhuang et al., 2023; Kumar et al., 2020). Besides, the state distribution of offline dataset $\mathcal{D}$ is usually unbalanced (Fu et al., 2020). As a result, offline to online finetuning faces severe **distribution shift** (Lee et al., 2022). Online finetuning should simultaneously improve the policy while eliminate the distribution shift. This entangled issue may lead to a sudden collapse in performance.

## 3 DOOF: DECOUPLED OFFLINE TO ONLINE FINETUNING FRAMEWORK

Previous offline to online finetuning methods directly finetune $\pi_{off}$ through online interaction (Nair et al., 2020; Nakamoto et al., 2024; Beeson & Montana, 2022). These approaches aim to eliminate distribution shift while simultaneously improve policy performance. However, there may be potential conflicts between these two objectives. Unlike previous work, our key insight lies in decoupling the elimination of distribution shift from the policy improvement within the model-based framework. concretely, our framework first eliminates the distribution shift through the online finetuning of dynamics $P_{\widehat{\mathcal{M}}}$ and then finetunes the policy $\pi_{off}$ in offline manner without online interaction.

We first theoretically decouple the offline to online finetuning into two stages and then reveal the relation between online dynamics finetuning and distribution shift elimination. Subsequently, we develop our algorithm **D**coupled **O**ffline to **O**nline **F**inetuning (DOOF) based on the MOPO and its uncertainty estimation. Last but not least, we discuss the advantages of this decoupled framework, especially on the interaction efficiency.

### 3.1 FINETUNING DYNAMICS MODEL THROUGH ONLINE INTERACTION

Within the context of model-based approaches, offline to online finetuning should minimize the gap between the $J\left(\pi_{off}, \widehat{\mathcal{M}}\right)$ and $J\left(\pi^*, \mathcal{M}\right)$. This gap encompasses not only a standard RL problem,

policy improvement, but also the error introduced by the inaccuracy of the dynamics model $P_{\widehat{\mathcal{M}}}$:

$$J\left(\pi_{\text{off}}, \widehat{\mathcal{M}}\right) - J\left(\pi^*, \mathcal{M}\right) = \underbrace{\left[J\left(\pi_{\text{off}}, \widehat{\mathcal{M}}\right) - J\left(\pi_{\text{off}}, \mathcal{M}\right)\right]}_{\text{error of the dynamics model}} + \underbrace{\left[J\left(\pi_{\text{off}}, \mathcal{M}\right) - J\left(\pi^*, \mathcal{M}\right)\right]}_{\text{policy optimization}}. \quad (3)$$

According to the above formulation, our model-based offline to online finetuning algorithm is divided into two phases. First, we aim to reduce the error of the dynamics model that can be viewed as the distributional shift in the context of offline to online finetuning. The relation between this error reduction (also the elimination of the distributional shift) and learning a more accurate $P_{\widehat{\mathcal{M}}}$ during online interaction will be revealed. Secondly, we leverage this refined dynamics model to improve the performance of $\pi_{\text{off}}$ in offline manner without further interaction with the environment.

**Step I: Elimination of distributional shift**  The relation between the first performance difference $J\left(\pi_{\text{off}}, \widehat{\mathcal{M}}\right) - J\left(\pi_{\text{off}}, \mathcal{M}\right)$ and the dynamics distance $d_{\text{TV}}\left(P_{\widehat{\mathcal{M}}}, P_{\mathcal{M}}\right)$ can be formulated as follows:

**Theorem 3.1** *Assume $\mathcal{M}$ and $\widehat{\mathcal{M}}$ are the MDPs with different transition dynamics $P_{\mathcal{M}}$ and $P_{\widehat{\mathcal{M}}}$ but the same reward function $r$. Then the performance difference $J\left(\pi_{\text{off}}, \widehat{\mathcal{M}}\right) - J\left(\pi_{\text{off}}, \mathcal{M}\right)$ holds:*

$$\left|J\left(\pi_{\text{off}}, \widehat{\mathcal{M}}\right) - J\left(\pi_{\text{off}}, \mathcal{M}\right)\right| \leq \frac{\gamma \cdot r_{\max}}{1 - \gamma} \mathbb{E}_{(s,a) \sim \rho_{\widehat{\mathcal{M}}}^{\pi_{\text{off}}}} \left[d_{\text{TV}}\left(P_{\widehat{\mathcal{M}}}, P_{\mathcal{M}}\right)\right]. \quad (4)$$

*The discounted unnormalized visitation frequencies $\rho_{\widehat{\mathcal{M}}}^{\pi_{\text{off}}}(s,a) = \pi_{\text{off}}(a|s) \cdot \sum_{t=0}^{T} \gamma^t P\left(s_t = s | \pi_{\text{off}}\right)$ and $P\left(s_t = s | \pi_{\text{off}}\right)$ represents the probability of the $t$-th state equals to $s$ in trajectories generated by policy $\pi_{\text{off}}$ and transition dynamics $P_{\widehat{\mathcal{M}}}$. The distance $d_{\text{TV}}\left(P_{\widehat{\mathcal{M}}}, P_{\mathcal{M}}\right)$ is the total variation distance and $|r| \leq r_{\max}$. More details about this theorem can be found in Appendix A.1.*

Furthermore, the dynamics distance $d_{\text{TV}}\left(P_{\widehat{\mathcal{M}}}, P_{\mathcal{M}}\right)$ can be bounded by the state-action frequency:

**Theorem 3.2** *Assume $\mathcal{P} = \{P_{\mathcal{M}} : \mathcal{S} \times \mathcal{A} \to \mathcal{S}\}$ and $|\mathcal{P}| < \infty$. Given an exact state-action pair $(s,a)$ exists in $\mathcal{D}$ with $\mathcal{D}_{s,a} = \{s_t, a_t, s_{t+1}\}_{s_t=s, a_t=a}$ and $n(s,a) = |\mathcal{D}_{s,a}|$. For $\delta \in (0,1)$ the dynamics model $P_{\widehat{\mathcal{M}}}$ learned by Equation 1 satisfies*

$$d_{\text{TV}}\left(P_{\widehat{\mathcal{M}}}, P_{\mathcal{M}}\right) \leq \sqrt{\frac{2 \log\left(|\mathcal{P}|/\delta\right)}{n(s,a)}}, \quad (5)$$

*with the probability at least $1 - \delta$. More details about this theorem can be found in Appendix A.2.*

**Discussion of Theorem 3.1 and Theorem 3.2**  Directly combining the two aforementioned theorems, we can derive the following inequality with the constant $C = \frac{\gamma \cdot r_{\max}}{1-\gamma} \sqrt{2 \log\left(|\mathcal{P}|/\delta\right)}$:

$$\frac{1}{C} \left|J\left(\pi_{\text{off}}, \widehat{\mathcal{M}}\right) - J\left(\pi_{\text{off}}, \mathcal{M}\right)\right| \leq \mathbb{E}_{(s,a) \sim \rho_{\widehat{\mathcal{M}}}^{\pi_{\text{off}}}} \left[\frac{1}{\sqrt{n(s,a)}}\right]$$

$$\leq \underbrace{\mathbb{E}_{(s_{\mathcal{D}}, a_{\mathcal{D}})} \left[\frac{1}{\sqrt{n(s,a)}}\right]}_{\text{a) offline dataset}} + \underbrace{\mathbb{E}_{(s_{\mathcal{D}}, a_{\neg\mathcal{D}})} \left[\frac{1}{\sqrt{n(s,a)}}\right]}_{\text{b) conservatism on OOD actions}} + \underbrace{\mathbb{E}_{(s_{\neg\mathcal{D}}, a)} \left[\frac{1}{\sqrt{n(s,a)}}\right]}_{\text{c) unbalanced state distribution}}. \quad (6)$$

On the right-hand side of the inequality, the entire state-action space is divided into three parts based on whether the state or action belongs to the offline dataset $\mathcal{D}$. Obviously, for the estimated dynamics model $P_{\widehat{\mathcal{M}}}$ trained using the offline dataset $\mathcal{D}$, the inequality $n(s_{\mathcal{D}}, a_{\mathcal{D}}) \ll n(s_{\mathcal{D}}, a_{\neg\mathcal{D}}) \ll n(s_{\neg\mathcal{D}}, a)$ holds. This implies that the first term in 6 will be relatively small, while the latter two terms are comparatively larger, especially the third term. During the online interaction, we aim to collect more data to reduce this performance gap caused by the distributional shift. The more efficient approach is to collect b) $(s_{\mathcal{D}}, a_{\neg\mathcal{D}})$ and c) $(s_{\neg\mathcal{D}}, a)$, rather than a) $(s_{\mathcal{D}}, a_{\mathcal{D}})$.

We train a dynamics model on $\mathtt{WK-m}$ dataset[2] and plot the distribution of the total variation distance across a) $(s_{\mathcal{D}}, a_{\mathcal{D}})$, b) $(s_{\mathcal{D}}, a_{\neg\mathcal{D}})$, c) $(s_{\neg\mathcal{D}}, a)$ in Figure 1. It is evident that the distance satisfies the inequality $d_{\mathcal{F},(s_{\mathcal{D}},a_{\mathcal{D}})} \ll d_{\mathcal{F},(s_{\mathcal{D}},a_{\neg\mathcal{D}})} \ll d_{\mathcal{F},(s_{\neg\mathcal{D}},a)}$, which aligns with our intuitive deductions. It is also important to note that the horizontal axis represents the $\log$ of the distance, indicating that the significant distance differences across different state-action regions, with even orders of magnitude gaps. This suggests that to reduce the expected dynamics distance, online interaction should make the second and third become smaller.

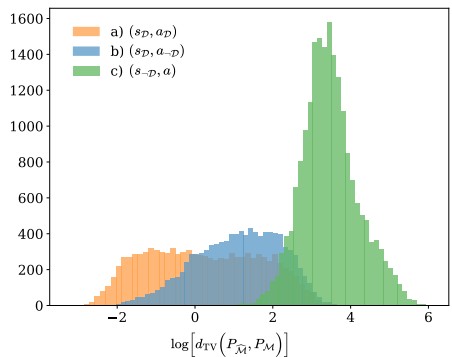

Figure 1: The distribution of total variation distance on $\mathtt{WK-m}$ dataset. Here we adopt the $\log$ on the distance due to the wide range.

**Step II: Offline policy improvement**    After online dynamics finetuning, one more accurate dynamics model $P^*_{\widehat{\mathcal{M}}}$ is obtained. Then we finetune the policy $\pi_{\text{off}}$ to get the final policy $\pi^*_{\text{off}}$ in offline manner, the same as the way of getting the offline pretrained policy $\pi_{\text{off}}$. Concretely, the offline manner represents optimizing the policy $\pi$ in conservative MDP

$$\widehat{\mathcal{M}}_u = \{\mathcal{S}, \mathcal{A}, r - \lambda_{\text{off}} \cdot u, P_{\widehat{\mathcal{M}}}, d_0, \gamma\}, \tag{7}$$

where the uncertainty $u(s,a) \geq d_{\text{TV}}\left(P_{\widehat{\mathcal{M}}}(s,a), P_{\mathcal{M}}(s,a)\right)$ for all $s \in \mathcal{S}, a \in \mathcal{A}$, is the upper bound of the dynamics distance. The performance of policy $\pi$ obtained from the offline policy optimization can be described using the following theorem.

**Theorem 3.3** *The performance of $\pi$, optimized in $\widehat{\mathcal{M}}_u = \{\mathcal{S}, \mathcal{A}, r - \lambda_{\text{off}} \cdot u, P_{\widehat{\mathcal{M}}}, d_0, \gamma\}$, satisfies*

$$J(\pi, \mathcal{M}) \geq J(\pi^*, \mathcal{M}) - 2\lambda_{\text{off}} \cdot \mathop{\mathbb{E}}_{(s,a)\sim\rho^{\pi^*}_{\widehat{\mathcal{M}}}} [u(s,a)]. \tag{8}$$

*Here $\pi^*$ is the optimal policy and more details about this theorem can be found in Appendix A.3.*

**Discussion of Theorem 3.3**    The performance $J(\pi, \mathcal{M})$ is affected by the uncertainty $u(s,a)$, also the distance between the true and estimated dynamics $d_{\mathcal{F}}\left(P_{\widehat{\mathcal{M}}}(s,a), P_{\mathcal{M}}(s,a)\right)$. After online dynamics pretraining, this distance becomes smaller. As a result, the offline finetuning policy $\pi^*_{\text{off}}$ is better $J(\pi^*_{\text{off}}, \mathcal{M}) \geq J(\pi_{\text{off}}, \mathcal{M})$.

### 3.2 Practical Implementation

Now we design a practical model-based offline to online finetuning framework called DOOF motivated by the above analysis. The framework has been summarized in Algorithm 1.

**Offline pretraining**    We first learn an ensemble dynamics $\left\{P^k_{\widehat{\mathcal{M}}}\right\}_{k=1}^K$ using Equation (1). All the uncertainty-based offline algorithms are applicable within our decoupled framework, such as MOPO (Yu et al., 2020), $\mathtt{count-MORL}$ (Kim & Oh, 2023) and MOBILE (Sun et al., 2023). We choose MOPO to verify our DOOF due to its simplicity and universality. Specifically, the uncertainty estimator is the maximum standard deviation of the learned dynamics models $u(s,a) = \max_{k=1}^K \left\|\Sigma^k_\phi(s,a)\right\|_{\text{F}}$. We denote the the policy obtained from offline pretraining as $\pi_{\text{off}}$.

**Online finetuning**    According to above analysis, finetuning dynamics model requires data, $(s_{\mathcal{D}}, a_{\neg\mathcal{D}})$ and $(s_{\neg\mathcal{D}}, a)$, that is out of the offline dataset distribution. If we interact with environment through offline pretrained policy $\pi_{\text{off}}$, the collected data mainly belongs to the distribution $(s_{\mathcal{D}}, a_{\mathcal{D}})$, which contributes little to learn a more accurate dynamics model. What we required is a more exploratory and optimistic policy, even if its performance is a little worse than $\pi_{\text{off}}$.

---

[2]Abbreviations of the datasets from D4RL are as follows: $\mathtt{halfcheetah} \rightarrow \mathtt{HC}$, $\mathtt{hopper} \rightarrow \mathtt{HP}$, $\mathtt{walker2d} \rightarrow \mathtt{WK}$, $\mathtt{Pen} \rightarrow \mathtt{P}$, $\mathtt{random} \rightarrow \mathtt{r}$, $\mathtt{medium} \rightarrow \mathtt{m}$, $\mathtt{medium-replay} \rightarrow \mathtt{mr}$, $\mathtt{medium-expert} \rightarrow \mathtt{me}$, $\mathtt{cloned} \rightarrow \mathtt{c}$, $\mathtt{human} \rightarrow \mathtt{h}$.

---

**Algorithm 1 D**ecoupled **O**ffline to **O**nline **F**inetuning (DOOF)

---

**Require:** Offline dataset $\mathcal{D}$; penalty coefficient $\lambda_{\text{off}}$; exploration coefficient $\lambda_{\text{on}}$.
    (**Offline pretraining**) Obtain the offline pretrained policy $\pi_{\text{off}}$ and offline dynamics model $P_{\widehat{\mathcal{M}}}$.
    (**Online finetuning**)
    1. Train uncertainty policy $\pi_u$ with modified reward $r_t + \lambda_{\text{on}} \cdot u\left(s_t, a_t\right)$.
    2. Collect data with $\pi_u$ in environment to get the real buffer $\mathcal{D}_{\text{on}}$.
    3. Finetune dynamics model $P_{\widehat{\mathcal{M}}}$ on dataset $\mathcal{D} \cup \mathcal{D}_{\text{on}}$.
    4. Finetune $\pi_{\text{off}}$ on dataset $\mathcal{D} \cup \mathcal{D}_{\text{on}}$ with finetuned dynamics model $P_{\widehat{\mathcal{M}}}^*$.

---

MOPO penalizes OOD state-action pairs using uncertainty $u(s, a)$. In Figure 2, we observe that the state-action region with greater distance $d_{\text{TV}}\left(P_{\widehat{\mathcal{M}}}, P_{\mathcal{M}}\right)$ also exhibits higher uncertainty $u(s, a)$. This implies that the uncertainty can assist in identifying which data points are more critical to be collected through online interaction. Therefore, we choose to use the uncertainty from MOPO as an extrinsic reward to train the policy $\pi_{\text{off}}$:

$$r_t^u\left(s_t, a_t\right) = r_t + \lambda_{\text{on}} \cdot u\left(s_t, a_t\right), \quad (9)$$

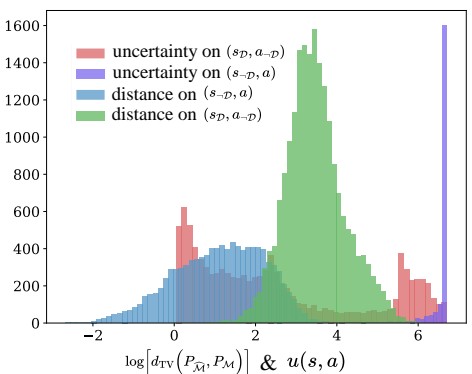

Figure 2: The distribution of total variation distance and uncertainty on `WK-m` dataset. The reason for the concentration of the uncertainty distribution on the right side is the clipping operation in the implementation.

here $\lambda_{\text{on}}$ is the exploration coefficient which determines the degree of uncertainty guidance. The selection of $\lambda_{\text{on}}$ is proportionally related to the offline penalty coefficient, where the ratio is $\lambda_{\text{on}} : \lambda_{\text{off}} = 0.25, 0.5, 1, 2$. We denote this modified policy as uncertainty policy $\pi_u$ and, for example, use $\pi_u(0.25)$ to represent the policy is trained with $\lambda_{\text{on}} : \lambda_{\text{off}} = 0.25$. Interacting with the environment, the uncertainty policy $\pi_u$ collects the online buffer $\mathcal{D}_{\text{on}}$. And similarly, $\mathcal{D}_{\text{on}}(0.25)$ indicates this online buffer is collected by $\pi_u(0.25)$. Then we finetune the dynamics model on data $\mathcal{D} \cup \mathcal{D}_{\text{on}}$:

$$P_{\widehat{\mathcal{M}}}^* = \arg \min_{P_{\widehat{\mathcal{M}}}} \mathbb{E}_{\mathcal{D} \cup \mathcal{D}_{\text{on}}}\left[-\log P_{\widehat{\mathcal{M}}}\left(s_{t+1} | s_t, a_t\right)\right]. \quad (10)$$

Finally, we run MOPO on dataset $\mathcal{D} \cup \mathcal{D}_{\text{on}}$ with the help of the finetuned dynamics model $P_{\widehat{\mathcal{M}}}^*$ to finetune the offline pretrained policy $\pi_{\text{off}}$ without the need of further online interaction.

### 3.3 Discussion and Advantages

The online finetuning stage of DOOF is actually the **supervised** dynamics model learning, which is significantly more straightforward than RL. To recover to the true transition dynamics on $(s_t, a_t)$, the dynamics model $P_{\widehat{\mathcal{M}}}$ only requires data on $(s_t, a_t)$. In contrast, RL problem naturally relies on more diverse data distribution. Taking the Q-function as an example, to learn the optimal $Q(s_t, a_t)$, it is not only data on $(s_t, a_t)$ is required, but also $(s_{t+1}, a_{t+1}, s_{t+2}, a_{t+2}, \cdots)$. This is one of the reasons why our framework boasts high interaction efficiency. Besides, the online buffer collected by the uncertainty policy tends to include state-action pairs with large uncertainty. This data distribution is precisely the region where the dynamics is less accurate, and also greatly contributes to the elimination of distribution shift.

Our policy finetuning is conducted in an offline manner without online interaction. If the dynamics is accurate enough, the policy is also equivalent to interacting with the environment. This is the inherent advantage of a model-based framework, which can generate a broader distribution with a small amount of training data. And this property further enhances the data efficiency. Moreover, the policy training stage remains conservative, which can prevent sudden performance drop.

## 4 RELATED WORK

**Offline reinforcement learning** Mainstream offline model-free methods mainly contains two categories. One is policy constraint, which constrains the learned policy close to the behavior policy based on different "closeness" such as batch constrained (Fujimoto et al., 2019), KL divergence (Wu et al., 2019), MMD distance (Kumar et al., 2019), MSE constraint (Fujimoto & Gu, 2021) and TV distance (Zhuang et al., 2023). The other is value regularization, which regularizes the value function from overestimation on OOD state-action pairs (Kumar et al., 2020; Kostrikov et al., 2021a; Bai et al., 2022). Besides, Decision Transformer (DT) (Chen et al., 2021) directly maximizes the action likelihood, which opens up a new paradigm called sequence modeling. And **Rein*for*mer** (Zhuang et al., 2024) further propose the max-return sequence modeling. Based on the conservatism on different components, offline model-based RL methods derived from MBPO (Janner et al., 2019) are divided into the following three categories: MOPO (Yu et al., 2020; Lu et al., 2021) and MOReL (Kidambi et al., 2020) propose to penalize the reward function by the uncertainty of the learned dynamics models and MOBILE (Sun et al., 2023) theoretically conducts uncertainty quantification through the inconsistency of Bellman estimations. COMBO (Yu et al., 2021) trains a conservative Q-function based on CQL (Kumar et al., 2020). RAMBO (Rigter et al., 2022) and ARMOR (Bhardwaj et al., 2024) incorporates conservatism by modifying the transition dynamics.

**Offline to online finetuning** Offline to online finetuning aims to further improve the policy using the offline pretrained policy as initialization. Offline to online finetuning encompasses two issues to be solved: distribution shift elimination (DSE) and policy improvement (PI). The approaches to dealing with the distribution shift can be roughly divided into two classes. One class tries to design less conservative algorithms (Lyu et al., 2022; Nakamoto et al., 2024; Kostrikov et al., 2021b; Wu et al., 2022), aiming to weaken the impact of distribution shift when online finetuning. Another class applies additional constraint (Beeson & Montana, 2022; Lee et al., 2022; Nair et al., 2020) to solve the distribution shift. Although MOORe (Mao et al., 2022), MOTO (Rafailov et al., 2023), and FOWM (Feng et al., 2023) are online finetuning algorithms within the model-based framework, they also directly finetunes the policy rather than decouples the dynamics learning from policy improvement like DOOF. A comparative analysis between DOOF and other offline to online finetuning methods, including the model-free algorithms (IQL (Kostrikov et al., 2021b), Cal-QL (Nakamoto et al., 2024), CQL (Kumar et al., 2020), SPOT (Wu et al., 2022), PEX (Zhang et al., 2023)) and model-based baselines such as MOORe (Mao et al., 2022) and FOWM (Feng et al., 2023), was conducted as listed in Table 1. The table 1 outlines the problems each algorithm attempts to address at each phase. Only DOOF achieves decoupling of these two issues, while the others simultaneously tackle the intertwined issues of distribution shift elimination and policy improvement.

Table 1: Problem to solve at each phase

| Methods | Online Finetuning | Offline Finetuning |
|---|---|---|
| IQL, Cal-QL, CQL, SPOT, PEX | Distribution Shift + Policy Improvement | No |
| MOORe | Distribution Shift + Policy Improvement | No |
| FOWM | Distribution Shift + Policy Improvement | No |
| DOOF (ours) | Distribution Shift | Policy Improvement |

## 5 EXPERIMENTS

We conduct an extensive and rigorous evaluation on our DOOF: **D**odel-based **O**ffline to **O**nline **F**inetuning using classical offline datasets from D4RL (Fu et al., 2020). Our experiments are organized in accordance with the algorithmic workflow:

- Uncertainty policy training: We evaluate the performance of uncertainty policy $\pi_u$ and illustrate the distribution of the collected online buffer $\mathcal{D}_{\text{on}}(\pi_u)$. The impact of exploration coefficient $\lambda_{\text{on}}$ on dynamics finetuning and policy finetuning is reserved for the latter sections.

- Online dynamics finetuning: We focus on the total variational distance between the finetuned dynamics model $P^*_{\widehat{\mathcal{M}}}$ and the actual environment, and the distance change before and after finetuning. We also analyze the influence of different $\lambda_{\text{on}}$ on above distance challenge.

- Offline policy finetuning: We plot the typical training curves to analyze the characteristics of DOOF and the impact of $\lambda_{\text{on}}$. We also demonstrate the exceptional interactive efficiency.

## 5.1 UNCERTAINTY POLICY TRAINING

The uncertainty policy $\pi_u$ determines the distribution of collected online buffer $\mathcal{D}_{\text{on}}(\pi_u)$, which directly affects the online dynamics fine-tuning and offline policy finetuning. When training $\pi_u$, we adopt four exploration coefficients $\lambda_{\text{on}}$ with ratio $\lambda_{\text{on}} : \lambda_{\text{off}} = 0.25, 0.5, 1, 2$. We first compare the performance between the offline pretrained policy $\pi_{\text{off}}$ and $\pi_u$.

In Figure 3, we illustrate the percentage of performance change $\left( \frac{J(\pi_u)}{J(\pi_{\text{off}})} - 1 \right) \times 100\%$ on ratio $0.25, 0.5, 1, 2$. Some datasets, such as `HP-m`, exhibits a noticeable performance decline. This is attributed to the uncertainty policy training that alters the inherent conservatism. But surprisingly, the `P-c` dataset achieves performance improvement through simple uncertainty training.

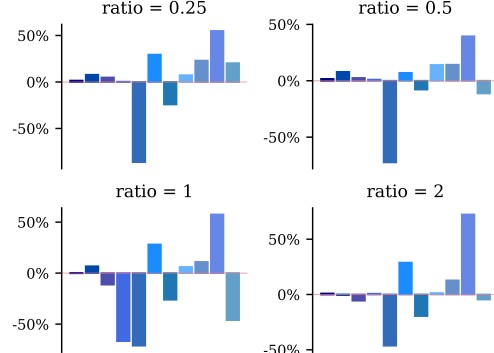

Figure 3: This figure depicts the percentage change on performance of the uncertainty policy $\pi_u$ compared to the offline pretrained policy $\pi_{\text{off}}$, where $\pi_u$ is trained with different exploration coefficients $\lambda_{\text{on}} : \lambda_{\text{off}} = 0.25, 0.5, 1, 2$. These datasets are `HC-r`, `HC-m`, `HC-mr`, `HP-r`, `HP-m`, `HP-mr`, `WK-r`, `WK-m`, `WK-mr`, `P-c`, `P-h`.

Next, we evaluate the distribution of online buffer $\mathcal{D}_{\text{on}}(\pi_u)$ collected by the uncertainty policy $\pi_u$. Ideally, the collected data distribution should consist of state-action regions unseen by the offline pretrained dynamics $P_{\widehat{\mathcal{M}}}$, that is, the distance $d_{\text{TV}}\left(P_{\widehat{\mathcal{M}}}, P_{\mathcal{M}}\right)$ on online buffer $\mathcal{D}_{\text{on}}(\pi_u)$ should be larger. In the left two figures from Figure 4, it can be observed that the distance on $\mathcal{D}$ is indeed minimal, while the distance of $\mathcal{D}_{\text{on}}(\pi_u)$ is substantially large. This suggests that the dynamics training of on $\mathcal{D}_{\text{on}}(\pi_u)$ could eliminate the distribution shift effectively.

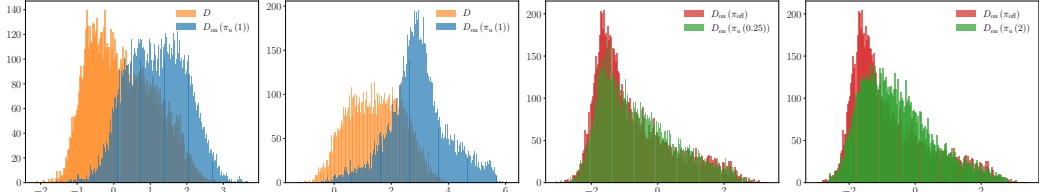

Figure 4: The left two figures illustrate the distribution of $\log\left[d_{\text{TV}}\left(P_{\widehat{\mathcal{M}}}, P_{\mathcal{M}}\right)\right]$ on the offline dataset $\mathcal{D}$ and the online buffer $\mathcal{D}_{\text{on}}$ collected by the uncertainty policy $\pi_u(1)$ trained with $\lambda_{\text{on}} = \lambda_{\text{off}}$, across datasets `HC-r` and `WK-r`. The third figure depicts the distance distribution on online buffer collected by the offline pretrained policy $\pi_{\text{off}}$ and uncertainty policy $\pi_u(0.25)$ trained with $\lambda_{\text{on}} = 0.25\lambda_{\text{off}}$ on `WK-mr`. While the last one is similar but the uncertainty policy is $\pi_u(0.25)$ on `WK-mr`.

In the right two figures from Figure 4, the distribution on online buffer collected by the offline pretrained policy $\pi_{\text{off}}$ and the uncertainty policy $\pi_u$ are almost overlap. This is because the $\pi_u$ is finetuned based on $\pi_{\text{off}}$. But these subtle differences actually have a significant impact on the performance of the final offline policy finetuning. We will further discuss in detail in Section 5.3.

## 5.2 ONLINE DYNAMICS FINETUNING

Table 2: The total variation distance between the true and estimated dynamics on state-action pairs uniformly sampled from the true MDP. $d_{\text{TV}}\left(P_{\widehat{\mathcal{M}}}, P_{\mathcal{M}}\right)$ represents the distance before online dynamics finetuning, also the offline pretrained dynamics model. $d_{\text{TV}}^{(\text{ratio})}\left(P_{\widehat{\mathcal{M}}}^*, P_{\mathcal{M}}\right)$ is the distance after finetuning, using the online buffer collected by the $\pi_u$ trained with $\lambda_{\text{on}} : \lambda_{\text{off}} = 0.25, 0.5, 1, 2$.

| Datasets | $d_{\text{TV}}(P_{\widehat{\mathcal{M}}}, P_{\mathcal{M}})$ | | $d_{\text{TV}}^{(0.25)}\left(P_{\widehat{\mathcal{M}}}^*, P_{\mathcal{M}}\right)$ | | $d_{\text{TV}}^{(0.5)}\left(P_{\widehat{\mathcal{M}}}^*, P_{\mathcal{M}}\right)$ | | $d_{\text{TV}}^{(1)}\left(P_{\widehat{\mathcal{M}}}^*, P_{\mathcal{M}}\right)$ | | $d_{\text{TV}}^{(2)}\left(P_{\widehat{\mathcal{M}}}^*, P_{\mathcal{M}}\right)$ | |
|---|---|---|---|---|---|---|---|---|---|---|
| | mean | max | mean | max | mean | max | mean | max | mean | max |
| HC-r | 118.123 | 1223.240 | 119.155 | 1294.893 | **109.950** | **1170.111** | 120.915 | 2076.364 | 119.825 | 1456.900 |
| HC-m | 82.339 | 910.501 | **63.708** | 969.096 | **67.705** | 1169.591 | **67.813** | 1029.322 | **67.565** | 1116.883 |
| HC-mr | 121.772 | 2445.304 | **93.484** | **1951.653** | **78.373** | **1500.777** | 89.847 | 1894.1079 | 89.317 | 1641.784 |
| HP-r | 39.970 | 319.989 | 44.888 | 425.438 | 56.680 | 547.203 | 56.527 | 629.646 | 56.668 | 672.624 |
| HP-m | 874.378 | 5567.265 | 1156.229 | 6678.457 | 950.298 | **5421.856** | **766.162** | 5719.151 | **765.820** | 5757.906 |
| HP-mr | 135.835 | 824.546 | **116.199** | 864.601 | **112.370** | **688.462** | 105.852 | 1401.446 | **105.176** | 1177.113 |
| WK-r | 211.203 | 2073.774 | **118.911** | **1149.800** | 92.696 | 793.019 | 145.021 | 1156.193 | 144.939 | 1234.787 |
| WK-m | 173.784 | 1018.120 | **156.550** | 973.955 | 157.137 | 909.678 | 149.325 | 988.611 | 148.906 | 997.185 |
| WK-mr | 164.754 | 1505.594 | **137.658** | **892.306** | 148.490 | 910.916 | 131.005 | 1104.575 | 130.517 | 1765.076 |

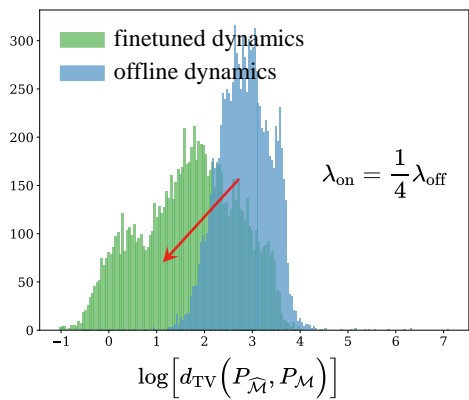

$$\lambda_{\text{on}} = \frac{1}{4}\lambda_{\text{off}}$$

$$\log\left[d_{\text{TV}}\left(P_{\widehat{\mathcal{M}}}, P_{\mathcal{M}}\right)\right]$$

Figure 5: This figure presents the distance distribution of the dynamics model pretrained by dataset HC-r on dataset HC-me before and after online dynamics finetuning. The dynamics model is finetuned using online buffer $\mathcal{D}_{\text{on}}(\pi_u(0.25))$.

Online dynamics finetuning aims to eliminate distribution shift by learning a more accurate dynamics model $P^*_{\widehat{\mathcal{M}}}$. The total variation distance of the dynamics model from the true environment on a relatively uniform state-action distribution can reflect the extent of distribution shift. The results are summarized in Table 2.

From the Table 2, we observe that the online dynamics finetuning significantly reduces the distance $d_{\text{TV}}\left(P_{\widehat{\mathcal{M}}}, P_{\mathcal{M}}\right)$ between the real environment on the fake buffer, except for HP-r with severely biased distribution. Moreover, dynamics finetuning demonstrates relative robustness with respect to the exploration coefficient $\lambda_{\text{on}}$.

In figure 5, we observe that distance distribution $d_{\text{TV}}\left(P_{\widehat{\mathcal{M}}}, P_{\mathcal{M}}\right)$, where $P_{\widehat{\mathcal{M}}}$ is pretrained on HC-r, exhibit long-tail effect on HC-me. This may be attributed to that the dynamics model learned from low-quality datasets is blind on the high-return regions. Through finetuning, not only the distance has been significantly reduced, but the long-tail distribution is also mitigated.

## 5.3 OFFLINE POLICY FINETUNING

We first demonstrate the superior finetuning performance and exceptional data efficiency of our framework DOOF. All the offline-to-online algorithms directly finetune the policy during the online interaction, hence they plot the policy training curves for comparison. In DOOF, the online interaction and policy finetuning are two decoupled stages, making this comparison method inapplicable. As a result, we compare our proposed DOOF with baselines using the following approaches:

- DOOF first online interacts 10k steps to finetune the dynamics model and then use this online finetuned dynamics to finetune the policy with 300k gradient steps in offline manner (OFF-300K);
- For baselines, including model-free IQL (Kostrikov et al., 2021b), Cal-QL (Nakamoto et al., 2024), CQL (Kumar et al., 2020), SPOT (Wu et al., 2022), PEX (Zhang et al., 2023) and model-based FOWM (Feng et al., 2023), they first finetune the policy using 10k online interaction steps (ON-10K) and then finetune the policy 300k gradient steps without online interaction (OFF-300K).

Table 3: The online (ON-10K) and offline (OFF-300K) finetuning results over 3 seeds. The gray results means the performance has declined after finetuning and the **bold** results represent the best.

| | DOOF | | IQL | | | CaL-QL | | | CQL | | |
|---|---|---|---|---|---|---|---|---|---|---|---|
| | | OFF-300K | | ON-10K | OFF-300K | | ON-10K | OFF-300K | | ON-10K | OFF-300K |
| HC-r | 37.5 | **54.9**(+17.4) | 14.9 | 14.2(−0.6) | 8.9(−6.0) | 25.1 | 11.2(−14.0) | 2.3(−22.9) | 23.7 | 12.3(−11.4) | 2.2(−21.5) |
| HC-m | 70.5 | **92.5**(+22.0) | 48.5 | 48.3(−0.2) | 49.2(+0.8) | 47.6 | 48.6(+1.0) | 49.1(+1.5) | 46.6 | 46.9(+0.2) | 47.7(+1.1) |
| HC-mr | 69.2 | **88.3**(+19.1) | 44.3 | 44.2(−0.0) | 45.3(+1.0) | 46.5 | 47.4(+0.9) | 47.4(+1.0) | 44.9 | 45.1(+0.2) | 46.0(+1.2) |
| HP-r | 32.0 | 32.4(+0.5) | 7.6 | 7.6(−0.0) | 8.17(+1.4) | 7.3 | 4.7(−2.6) | 2.0(−5.3) | 7.8 | 6.9(−1.0) | 8.6(+0.8) |
| HP-m | 68.3 | **108.4**(+40.1) | 57.3 | 58.8(+1.4) | 59.0(+1.6) | 56.8 | 72.7(+15.9) | 73.5(+16.7) | 62.3 | 65.4(+3.1) | 63.0(+0.7) |
| HP-mr | 82.6 | **107.3**(+24.7) | 97.2 | 95.7(−1.5) | 99.4(+2.2) | 97.7 | 97.3(−0.5) | 97.7(−0.1) | 94.1 | 95.0(+0.9) | 99.7(+5.7) |
| WK-r | 4.1 | **16.9**(+12.8) | 3.7 | 3.7(+0.0) | 4.2(+0.6) | 3.9 | 1.5(−2.4) | 0.2(−3.7) | −0.3 | −0.2(+0.0) | 1.6(+1.9) |
| WK-m | 77.7 | **93.8**(+16.1) | 83.2 | 80.8(−2.4) | 79.0(−4.2) | 83.6 | 83.6(+0.0) | 83.3(−0.3) | 81.3 | 83.6(+2.3) | 80.5(−0.8) |
| WK-mr | 69.6 | **97.8**(+28.2) | 77.1 | 78.1(+1.0) | 84.2(+7.1) | 79.4 | 87.2(+7.8) | 89.2(+9.8) | 72.1 | 82.2(+10.2) | 87.6(+15.5) |
| P-c | 48.9 | 54.2(+5.3) | 69.5 | 67.9(−1.6) | 68.9(−0.6) | −3.5 | −4.0(−0.6) | −4.8(−1.3) | −3.4 | −3.8(−0.4) | −4.8(−1.4) |
| P-h | 38.7 | **62.0**(+23.3) | 56.4 | 57.6(+1.2) | 60.2(+3.8) | 7.7 | −4.0(−11.6) | −3.4(−11.1) | −3.4 | −3.5(−0.2) | −4.4(−1.1) |

| | DOOF | | SPOT | | | PEX | | | FOWM | | |
|---|---|---|---|---|---|---|---|---|---|---|---|
| | | OFF-300K | | ON-10K | OFF-300K | | ON-10K | OFF-300K | | ON-10K | OFF-300K |
| HC-r | 37.5 | **54.9**(+17.4) | 8.0 | 7.8(−0.2) | 5.1(−2.8) | 15.7 | 16.4(+0.7) | 8.6(−7.1) | 15.4 | 20.4(+5.0) | 20.3(+4.9) |
| HC-m | 70.5 | **92.5**(+22.0) | 45.4 | 46.2(+0.8) | 45.1(−0.3) | 48.3 | 49.5(+1.2) | 46.0(−2.3) | 44.0 | 45.4(+1.4) | 42.9(−1.1) |
| HC-mr | 69.2 | **88.3**(+19.1) | 43.8 | 43.1(−0.7) | 43.2(−0.6) | 44.7 | 45.7(+1.0) | 44.2(−0.5) | 47.3 | 49.8(+2.5) | 49.7(+2.3) |
| HP-r | 31.9 | 32.4(+0.5) | 8.5 | 9.9(+1.4) | **32.3**(+23.8) | 8.4 | 7.7(−0.7) | 7.8(−0.6) | 9.2 | 9.5(+0.3) | 9.5(+0.3) |
| HP-m | 68.3 | **108.4**(+40.1) | 57.8 | 59.0(+1.2) | 60.9(+3.1) | 59.2 | 19.4(−39.8) | 56.6(−2.6) | 48.5 | 64.5(+16.0) | 52.5(+4.0) |
| HP-mr | 82.6 | **107.3**(+24.7) | 74.2 | 47.9(−26.4) | 85.5(+11.2) | 78.3 | 102.3(+24.0) | 73.1(−5.2) | 93.0 | 88.0(−5.0) | 99.3(+6.3) |
| WK-r | 4.1 | **16.9**(+12.8) | 1.5 | 6.1(+4.6) | 5.7(+4.2) | 8.8 | 8.6(−0.2) | 9.1(+0.3) | 4.2 | 5.6(+1.4) | 5.5(+1.2) |
| WK-m | 77.7 | **93.8**(+16.1) | 81.7 | 81.0(−0.8) | 82.5(+0.8) | 72.1 | 56.8(−15.3) | 81.9(+9.8) | 0.6 | 40.0(+39.4) | 27.3(+26.8) |
| WK-mr | 69.6 | **97.8**(+28.2) | 81.1 | 81.0(−0.0) | 81.8(+0.8) | 71.3 | 61.8(−9.5) | 59.7(−11.6) | 36.0 | 54.7(+18.7) | 45.3(+9.3) |
| P-c | 48.9 | 54.2(+5.3) | 2.5 | 15.5(+13.0) | 22.5(+20.0) | 33.8 | 29.8(−4.0) | 46.7(+12.9) | −2.8 | 51.0(+53.8) | **57.3**(+60.1) |
| P-h | 38.7 | **62.0**(+23.3) | 25.9 | 13.7(−12.2) | 18.2(−7.6) | 50.1 | 59.0(+8.9) | 70.6(+20.5) | −0.0 | 1.9(+1.9) | 8.3(+8.3) |

Our performance improvement is highly significant, approaching and even achieving optimal policy levels. What's more notable is the interaction efficiency; we have achieved remarkable effects with only 10k online interaction steps. In addition to the model-based approach, the decoupled framework and the uncertainty guided data collection to quickly eliminate distribution shift are also crucial.

We further plot several representative curves to illustrate the stable training performance and the relationship between performance and the dynamics distance. In Figure 6a, if we set the policy that collects the online buffer as $\pi_{\text{off}}$ (the blue bold curve), the offline finetuning performance may decline. While the performance with uncertainty policy $\pi_u(2)$ is best. This suggests that the policy guided by uncertainty is more conducive to collecting data that is effective for finetuning the dynamics model. In the third figure from Figure 4, the distance distribution is similar, which means the minor distribution differences can have a significant impact on the final policy finetuning performance. In Figure 6b, the stability of offline finetuning curves obtained by different data collection policy varies. The offline pretrained policy is $\pi_{\text{off}}$ is relatively poor, while other uncertainty policy is more stable. This indicates the distribution difference in Figure 4 affects the stability heavily.

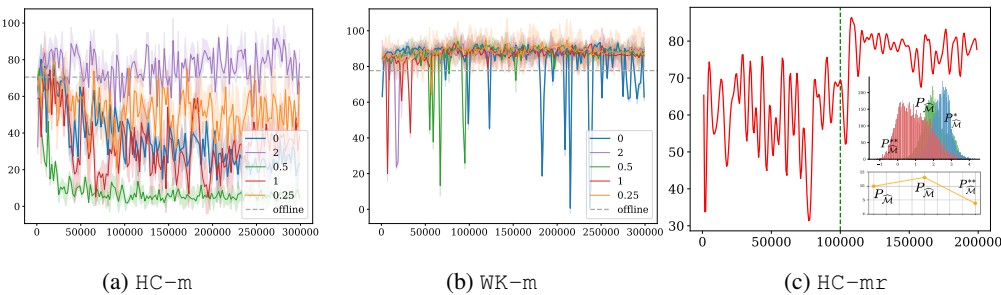

(a) HC-m                                    (b) WK-m                                    (c) HC-mr

Figure 6: Figure 6a and 6b illustrate the performance curves during policy finetuning on HC-m and WK-m. Note that the horizontal axis represents the training steps, rather than the interaction steps. Different curves represents different exploration coefficient $\lambda_{\text{on}}$. The blue bold curve with label '0' is based on the offline pretrained policy $\pi_{\text{off}}$ rather than uncertainty policy. Figure 6c shows the curve that carries on twice online finetuning. The curve before the green dashed line represents the policy finetuning under finetuned dynamics model $P^*_{\widehat{\mathcal{M}}}$, while the other is the policy finetuning curves with $P^{**}_{\widehat{\mathcal{M}}}$, the dynamics further finetuned based on the $P^*_{\widehat{\mathcal{M}}}$. The two small plots in the lower right corner respectively illustrate the distribution of distances of three different dynamics $P_{\widehat{\mathcal{M}}}, P^*_{\widehat{\mathcal{M}}}, P^{**}_{\widehat{\mathcal{M}}}$, as well as the curve of distance mean. The distance is also calculated on its offline fake buffer.

In Figure 6c, we conduct another finetuning process after the the policy has been finetuned. Dynamics model $P^*_{\widehat{\mathcal{M}}}$ is obtained form the first online dynamics finetuning while $P^{**}_{\widehat{\mathcal{M}}}$ is further finetuned based on $P^*_{\widehat{\mathcal{M}}}$. The distance distributions of these two dynamics models between the true environment are shown in the lower right corner. It can be observed that after the first finetuning, the distance increases, which means this finetuning is a failure that corresponds to a decline in performance and severe fluctuations. After the second finetuning, the distance becomes lower than the offline pretrained dynamics $P_{\widehat{\mathcal{M}}}$, resulting in a noticeable improvement in performance.

# 6 CONCLUSION AND FUTURE WORK

In this work, we propose a decoupled model-based offline to online finetuning framework called DOOF. DOOF decouples the distribution shift elimination from the policy optimization in offline to online finetuning. Specifically, DOOF finetunes the dynamics model during online interaction to eliminate the distribution shift using the online buffer collected by the uncertainty guided policy. And then the policy is finetuned with the help of the finetuned dynamics in offline manner without further interaction. This decoupled framework not only ensures superior and stable performance but also boasts exceptional interaction efficiency. Overall, model-based offline algorithms lag behind model-free counterparts in terms of performance. Therefore, we intend to explore how to extend this decoupled offline to online finetuning framework to model-free algorithms by incorporating an additional trained dynamics model. This integration approach may be more conducive to unlocking the full potential of the decoupled framework.

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

# A   PROOF

## A.1   PROOF OF THEOREM 3.1 (YU ET AL., 2020)

Assume $\mathcal{M}$ and $\widehat{\mathcal{M}}$ are the MDPs with different transition dynamics $P_{\mathcal{M}}$ and $P_{\widehat{\mathcal{M}}}$ but the same reward function $|r| \leq r_{\max}$. According to telescoping lemma from Luo et al. (2018), the performance difference $J\left(\pi_{\text{off}}, \widehat{\mathcal{M}}\right) - J\left(\pi_{\text{off}}, \mathcal{M}\right)$ holds:

$$\left| J\left(\pi_{\text{off}}, \widehat{\mathcal{M}}\right) - J\left(\pi_{\text{off}}, \mathcal{M}\right) \right| \leq \gamma \mathop{\mathbb{E}}_{(s,a) \sim \rho_{\widehat{\mathcal{M}}}^{\pi_{\text{off}}}} \left[ d_{\mathcal{F}}\left(P_{\widehat{\mathcal{M}}}(s,a), P_{\mathcal{M}}(s,a)\right) \right]. \tag{11}$$

Here the state-action pairs comes from the discounted unnormalized visitation frequencies $\rho_{\widehat{\mathcal{M}}}^{\pi_{\text{off}}}(s,a) = \pi_{\text{off}}(a|s) \cdot \sum_{t=0}^{T} \gamma^t P\left(s_t = s|\pi_{\text{off}}\right)$ and $P\left(s_t = s|\pi_{\text{off}}\right)$ represents the probability of the $t$-th state equals to $s$ in trajectories generated by policy $\pi_{\text{off}}$ and transition dynamics $P_{\widehat{\mathcal{M}}}$. The distance $d_{\mathcal{F}}\left(P_{\widehat{\mathcal{M}}}(s,a), P_{\mathcal{M}}(s,a)\right) = \sup_{f \in \mathcal{F}} \left| \mathop{\mathbb{E}}_{s' \sim P_{\widehat{\mathcal{M}}}(s,a)}[f(s')] - \mathop{\mathbb{E}}_{s' \sim P_{\mathcal{M}}(s,a)}[f(s')] \right|$ with $d_{\mathcal{F}}$ is the is the integral probability metric (IPM) (Müller, 1997). IPMs are quite general (Sriperumbudur et al., 2009) and one special case is the total variational distance.

When $\mathcal{F} = \{f : \|f\|_\infty \leq 1\}$, $d_{\mathcal{F}}$ becomes the total variational distance $d_{\text{TV}}\left(P_{\widehat{\mathcal{M}}}, P_{\mathcal{M}}\right)$. Due to the bounded reward function $|r| \leq r_{\max}$, the $\|V^\pi\| \leq \sum_{t=0}^{\infty} \gamma^t r_{\max} = \frac{r_{\max}}{1-\gamma}$ holds. Then we have

$$\left| J\left(\pi_{\text{off}}, \widehat{\mathcal{M}}\right) - J\left(\pi_{\text{off}}, \mathcal{M}\right) \right| \leq \frac{\gamma \cdot r_{\max}}{1 - \gamma} \mathop{\mathbb{E}}_{(s,a) \sim \rho_{\widehat{\mathcal{M}}}^{\pi_{\text{off}}}} \left[ d_{\text{TV}}\left(P_{\widehat{\mathcal{M}}}, P_{\mathcal{M}}\right) \right]. \tag{12}$$

## A.2   PROOF OF THEOREM 3.2 (KIM & OH, 2023)

Assume $\mathcal{P} = \{P_{\mathcal{M}} : \mathcal{S} \times \mathcal{A} \to \mathcal{S}\}$ and $|\mathcal{P}| < \infty$. Given an exact state-action pair $(s, a)$ exists in $\mathcal{D}$ with $\mathcal{D}_{s,a} = \{s_t, a_t, s_{t+1}\}_{s_t=s, a_t=a}$ and $n(s,a) = |\mathcal{D}_{s,a}|$. By theorem 21 from Agarwal et al. (2020), the dynamics model $P_{\widehat{\mathcal{M}}}$ learned by Equation 1 satisfies

$$\mathbb{E}_{(s_t, a_t) \sim \mathcal{D}_{s,a}} \left[ d_{\text{TV}}\left(P_{\widehat{\mathcal{M}}}(s_t, a_t), P_{\mathcal{M}}(s_t, a_t)\right)^2 \right] \leq \frac{2 \log\left(\mathcal{M}/\delta\right)}{n\left(s, a\right)} \tag{13}$$

with the probability at least $1 - \delta$. We can directly bound the total variation distance between the estimated transition dynamics $P_{\widehat{\mathcal{M}}}$ and the true transition dynamics $P_{\mathcal{M}}$ due to the subset $\mathcal{D}_{s,a} = \{(s, a, s_{t+1})\}_{t=0}^{n(s,a)}$. Thus,

$$\mathbb{E}_{(s_t, a_t) \sim \mathcal{D}_{s,a}} \left[ d_{\text{TV}}\left(P_{\widehat{\mathcal{M}}}(s_t, a_t), P_{\mathcal{M}}(s_t, a_t)\right)^2 \right]$$

$$= \int \left[ P(s_i, a_i) d_{\text{TV}}\left(P_{\widehat{\mathcal{M}}}(s_t, a_t), P_{\mathcal{M}}(s_t, a_t)\right)^2 \right] \text{d}(s_i, a_i)$$

$$\geq \int \left[ P(s, a) d_{\text{TV}}\left(P_{\widehat{\mathcal{M}}}(s_t, a_t), P_{\mathcal{M}}(s_t, a_t)\right)^2 \right] \text{d}(s_i, a_i)$$

$$= d_{\text{TV}}\left(P_{\widehat{\mathcal{M}}}(s, a), P_{\mathcal{M}}(s, a)\right)^2. \tag{14}$$

Finally, we have

$$d_{\text{TV}}\left(P_{\widehat{\mathcal{M}}}, P_{\mathcal{M}}\right) \leq \sqrt{\frac{2 \log\left(\mathcal{M}/\delta\right)}{n\left(s, a\right)}}, \tag{15}$$

## A.3   PROOF OF THEOREM 3.3 (YU ET AL., 2020)

According to Theorem 4.3 from Yu et al. (2020), the policy optimized with uncertainty-based offline model-based algorithm $\widehat{\mathcal{M}}_u = \{\mathcal{S}, \mathcal{A}, r - \lambda_{\text{off}} \cdot u, P_{\widehat{\mathcal{M}}}, d_0, \gamma\}$ satisfies

$$J(\pi, \mathcal{M}) \geq \sup_{\hat{\pi}} \left[ J(\hat{\pi}, \mathcal{M}) - 2\lambda_{\text{off}} \cdot \mathop{\mathbb{E}}_{(s,a) \sim \rho_{\widehat{\mathcal{M}}}^{\hat{\pi}}} \left[ u(s, a) \right] \right]. \tag{16}$$

Assume $\pi^*$ is the optimal policy $\pi^* = \arg\max\limits_{\hat{\pi}} J(\hat{\pi}, \mathcal{M})$, then we directly have

$$J(\pi, \mathcal{M}) \geq J(\pi^*, \mathcal{M}) - 2\lambda_{\text{off}} \cdot \mathbb{E}_{(s,a)\sim\rho^{\pi^*}_{\widehat{\mathcal{M}}}}[u(s,a)]. \tag{17}$$

# B  IMPLEMENT DETAILS

## B.1  OFFLINE TRAINING

we firstly run MOPO on the environments to get offline trained dynamics model and policy. During this stage, We train an ensemble of 7 dynamics models, each model in the ensemble is represented as a 5-layer feedforward neural network with 400 hidden units on Adroit domain, but 4-layer feedforward neural network with 200 hidden units on other Gym domain. Once dynamics model training finished , we use it to rollout for generating data and helping training the policy. During model rollouts, we randomly pick 5 dynamics model from the 7 models. Finally, the mix batch consists of 5% from an offline dataset and the rest from dynamic rollouts, which is used to train the policy optimized by SAC.

## B.2  UNCERTAINTY POLICY TRAINING

We initialize the uncertainty policy as former offline pretrained policy. The uncertainty policy optimization is based on SAC, we sample a uncertainty dataset of 10000 transitions from offline dataset and modify the reward by adding the uncertainty which is determined by our online exploration coefficient and uncertainty value evaluated by offline trained dynamics model back to reward. For each update, we sample a mix batch of 256 transitions where 2.5% of them is from the offline dataset, 2.5% of them is from the modified uncertainty dataset, and 95% is from the synthetic dataset generated by the offline trained dynamics model. During dynamics model rollouts, half of the observations required for dynamics rollout comes from offline dataset and another half comes from the modified uncertainty dataset. We expect this training stage on the uncertainty policy will enhance its exploration.

## B.3  ONLINE DATA COLLECTING

we just utilize the trained uncertainty policy to act in environment for collecting online interacting data which was only used for finetuning the dynamics later without any further training. It finds out that very small interact steps setup on this stage can make significant impact on dynamics finetuning. Actually we just collect $10\,000$ online transitions, which is of great help to finetune the dynamics and enormously improve the policy performance.

## B.4  DYNAMICS FINETUNING

After online data collecting finished, we finetune the offline trained dynamics in original offline mode on both offline dataset and online collected dataset, for each update, the mix batch consists of half from offline dataset and another half from online collected dataset is used to train the dynamics model.

## B.5  POLICY FINETUNING

The policy to finetune was initialized offline pretrained policy. Once dynamics finetuning finished, it's used to assist in finetuning the policy in original offline mode with no need to interact with environment further, which fundamentally avoid sudden policy collapse because the inherent conservatism is retained and the algorithmic consistency is ensured. while finetuning the policy, for each update, we sample a mix batch size of 256 transitions consists of 2.5% from offline dataset and 97.5% from the synthetic dataset which is generated by the finetuned dynamics model. During dynamics models rollouts, half of the observations required to generate this synthetic dataset comes from offline dataset and another half comes from the online collected dataset.

## C    EXPERIMENT DETAILS

Our experiments is conducted based on the open source code base `https://github.com/yihaosun1124/OfflineRL-Kit`. we add four additional parts of code as described in section B. The code has been submitted as supplementary material. A single experiment requires approximately 40 hours of training on V100 during the offline pretraining phase, while the online phase only requires 4 to 5 hours.

### C.1    BENCHMARKS

we conduct experiments on D4RL benchmark, including Gym tasks(V2) and some Adroit tasks (V1), we choose several representative algorithms as baselines to show the priority of our algorithm, and implement the baseline experiments using flowing repositories:

- IQL, CQL, CAL-QL, and SPOT: `https://github.com/tinkoff-ai/CORL`
- PEX: `https://github.com/Haichao-Zhang/PEX`
- FOWM:`https://github.com/fyhMer/fowm`

we keep the origin code style of every code base, and just change the online finetuning steps and offline finetuning steps to keep the same as ours, we set the online interaction steps as 10 000 for all the baselines, and we modify the baselines based on their original code to conduct offline policy finetuning after online finetuning , and we set the offline finetuning steps as 300 000 on all baseline algorithms.

### C.2    HYPERPARAMETERS

Now we list the hyperparameters we have tuned for DOOF as follow.

**Exploratory coefficient $\lambda_{\text{on}}$.**    we tune the $\lambda_{\text{on}}$ in the range of $\lambda_{\text{on}} : \lambda_{\text{off}} = \{0.25, 0.5, 1, 2\}$. This hyperparameter has been discussed in section 5. we have listed this hyperparameter of different task in Table 5.

### C.3    TUNING FOR MOPO

All code parameters are default parameters in the code repository `https://github.com/yihaosun1124/OfflineRL-Kit`. While there are tasks that are not implemented in the repository, therefore we implemented these tasks and tuned these hyperparameters. Most of hyperparameters for MOPO are listed in Table 4.

Table 4: Hyperparameters of MOPO used in the D4RL datasets.

| Hyparameters | Value | Description |
|---|---|---|
| $lr_{\text{actor}}$ | 1e-4 (3e-5 on Adroit) | Learning rate of the actor network |
| $lr_{\text{critic}}$ | 3e-4 | Learning rate of the critic network |
| $lr_{\text{dynamics}}$ | 1e-3 (3e-4 on Adroit) | Learning rate of the actor network |
| $N$ | 2(3 on Adroit) | Number of hidden layers of actor and critic network |
| $N_{\text{dynamics}}$ | 4 | Number of hidden layers of dynamics network |
| $N_{\text{ensemble}}$ | 7 | Dynamics model ensemble size |
| Optimizer | Adam | Type of optimizer |
| $h$ | 256 | Number of hidden layer dimensions of actor and critic network |
| $h_{\text{dynamics}}$ | 200(400 on Adroit) | Number of hidden layer dimensions of dynamics network |
| $\gamma$ | 0.99 | Discount return |
| $K$ | 3000 | Number of training epochs |

**Rollout length $\mathcal{L}$.**    We perform short-horizon branch rollouts in MOPO. We tune $\mathcal{L}$ in the range of $\{1, 5, 10\}$ for Adroit tasks, and also for Gym tasks of random dataset, as listed in Table 5.

**Penalty coefficient $\lambda_{\text{off}}$.**    we tune $\lambda_{\text{off}}$ in the range of $\{0.25, 0.5, 0.75, 2.5, 5.0\}$ for Adroit tasks, and also for Gym tasks of random dataset, as listed in Table 5.

Table 5: Hyperparameters related to dynamics rollout of different tasks.

| Dataset | rollout length $\mathcal{L}$ | offline penalty coefficient $\lambda_{\text{off}}$ | online exploratory coefficient $\lambda_{\text{on}}$ |
|---|---|---|---|
| HC-r | 1 | 0.25 | $0.5\ (= 2\lambda_{\text{off}})$ |
| HC-m | 5 | 0.5 | $1.0\ (= 2\lambda_{\text{off}})$ |
| HC-mr | 5 | 0.5 | $1.0\ (= 2\lambda_{\text{off}})$ |
| HP-r | 1 | 5.0 | $10.0\ (= 2\lambda_{\text{off}})$ |
| HP-m | 5 | 5.0 | $2.5\ (= 0.5\lambda_{\text{off}})$ |
| HP-mr | 5 | 2.5 | $5.0\ (= 2\lambda_{\text{off}})$ |
| WK-r | 1 | 0.75 | $1.5\ (= 2\lambda_{\text{off}})$ |
| WK-m | 5 | 0.5 | $0.125\ (= 0.25\lambda_{\text{off}})$ |
| WK-mr | 1 | 2.5 | $1.25\ (= 0.5\lambda_{\text{off}})$ |
| P-c | 1 | 2.5 | $1.25\ (= 0.5\lambda_{\text{off}})$ |
| P-h | 1 | 0.5 | $0.25\ (= 0.5\lambda_{\text{off}})$ |

Table 6: The offline policy evaluations score mean, standard deviation and offline(OFF-300K) fine-tuning policy evaluations score mean, standard deviation.

| Dataset | DOOF | |
|---|---|---|
| | | OFF-300K |
| HC-r | $37.52 \pm (\ 2.78)$ | $54.89 \pm (\ 3.09)$ |
| HC-m | $70.50 \pm (\ 6.12)$ | $92.48 \pm (\ 5.58)$ |
| HC-mr | $69.20 \pm (\ 1.62)$ | $88.28 \pm (\ 3.17)$ |
| HP-r | $31.90 \pm (\ 0.66)$ | $32.39 \pm (\ 0.64)$ |
| HP-m | $68.28 \pm (24.46)$ | $108.40 \pm (\ 1.05)$ |
| HP-mr | $82.61 \pm (30.76)$ | $107.34 \pm (\ 0.81)$ |
| WK-r | $4.05 \pm (\ 0.34)$ | $16.85 \pm (\ 6.75)$ |
| WK-m | $77.67 \pm (15.18)$ | $93.77 \pm (\ 1.27)$ |
| WK-mr | $69.60 \pm (11.60)$ | $97.83 \pm (\ 0.80)$ |
| P-c | $48.90 \pm (11.89)$ | $54.19 \pm (\ 6.78)$ |
| P-h | $38.66 \pm (24.21)$ | $61.99 \pm (11.67)$ |

# D DISTANCE AND UNCERTAINTY DISTRIBUTION RESULTS

## D.1 UNCERTAINTY AND DISTANCE OF OFFLINE DYNAMICS

As discussed in subsection 3.1, the same phenomenon about distance between offline dynamics and true environment dynamics was observed in other tasks under different dataset, as listed in Fig 7.The distance satifies the inequality $d_{\mathcal{F},(s_{\mathcal{D}},a_{\mathcal{D}})} \ll d_{\mathcal{F},(s_{\mathcal{D}},a_{\neg\mathcal{D}})} \ll d_{\mathcal{F},(s_{\neg\mathcal{D}},a)}$ on all tasks.

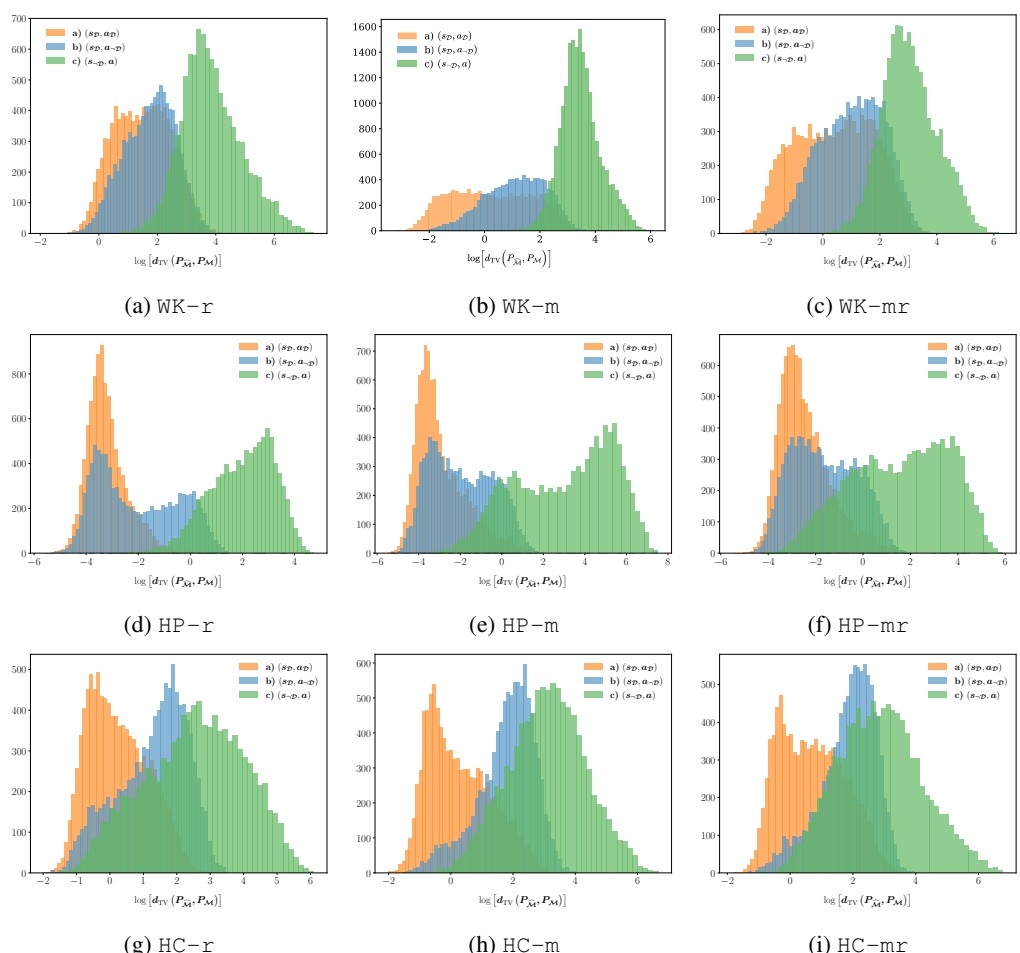

Figure 7: The distribution of total variation distance on different dataset.

As discussed in subsection 3.2, the same phenomenon about distance and uncertainty in other tasks under different dataset was observed. We have displayed the distribution of total variation distance and uncertainty on different task and datasets in Fig 8. In all tasks under different dataset, the state-action region with greater distance $d_{\mathrm{TV}}\left(P_{\widehat{\mathcal{M}}}, P_{\mathcal{M}}\right)$ also exhibits higher uncertainty $u(s,a)$ which obviously implies that the uncertainty can assist in identifying which data points are more critical to be collected through online interaction.

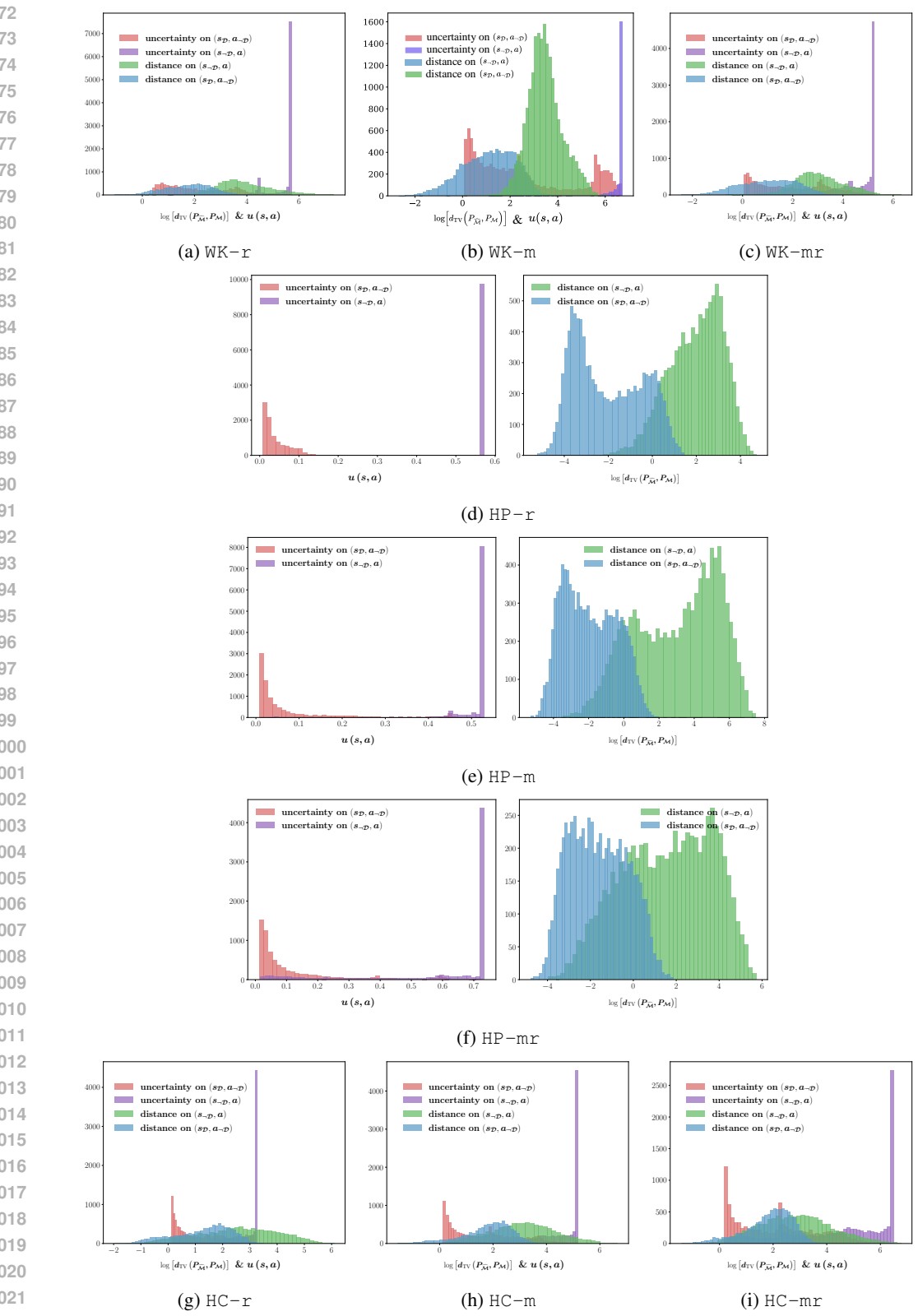

Figure 8: The distribution of total variation distance and uncertainty on different dataset. The reason for the concentration of the uncertainty distribution is the clipping operation in the implementation. On the HP tasks, the scale ranges of uncertainty and distance differ significantly, so it's displayed separately.

## D.2 DISTANCE OF FINETUNED DYNAMICS

Table 7: The total variation distance between the true environment and dynamics model on the fake buffer. $d_{\text{TV}}\left(P_{\widehat{\mathcal{M}}}, P_{\mathcal{M}}\right)$ represents the distance before online dynamics finetuning, also the offline pretrained dynamics model. $d_{\text{TV}}^{(\text{ratio})}\left(P_{\widehat{\mathcal{M}}}^*, P_{\mathcal{M}}\right)$ is the distance after finetuning, using the online buffer collected by the uncertainty policy $\pi_u$ trained with $\lambda_{\text{on}} : \lambda_{\text{off}} = 0.25, 0.5, 1, 2$.

| Datasets | $d_{\text{TV}}\left(P_{\widehat{\mathcal{M}}}, P_{\mathcal{M}}\right)$ | | $d_{\text{TV}}^{(0.25)}\left(P_{\widehat{\mathcal{M}}}^*, P_{\mathcal{M}}\right)$ | | $d_{\text{TV}}^{(0.5)}\left(P_{\widehat{\mathcal{M}}}^*, P_{\mathcal{M}}\right)$ | | $d_{\text{TV}}^{(1)}\left(P_{\widehat{\mathcal{M}}}^*, P_{\mathcal{M}}\right)$ | | $d_{\text{TV}}^{(2)}\left(P_{\widehat{\mathcal{M}}}^*, P_{\mathcal{M}}\right)$ | |
|---|---|---|---|---|---|---|---|---|---|---|
| | mean | max | mean | max | mean | max | mean | max | mean | max |
| HC-r | 19.705 | 93.315 | **2.911** | **29.336** | **2.962** | **29.864** | **2.618** | **32.465** | **2.817** | **36.587** |
| HC-m | 16.305 | 86.622 | **3.800** | **59.283** | **6.356** | **67.574** | **6.918** | **68.005** | **5.450** | 94.405 |
| HC-mr | 15.406 | 63.238 | **6.375** | **51.478** | **3.616** | 63.864 | **9.342** | **58.687** | **4.665** | **57.970** |
| HP-r | 0.893 | 6.359 | **0.199** | 31.883 | **0.231** | **3.719** | 1.477 | 62.536 | **0.064** | **2.563** |
| HP-m | 0.962 | 8.770 | **0.172** | **6.119** | **0.436** | **7.290** | **0.430** | **7.521** | **0.595** | **6.702** |
| HP-mr | 2.371 | 22.685 | **0.216** | **6.159** | **0.674** | **9.366** | **0.558** | **7.228** | **0.343** | **6.705** |
| WK-r | 25.833 | 244.954 | **11.848** | **195.372** | **11.390** | **180.553** | **12.553** | **218.340** | **12.511** | **114.747** |
| WK-m | 7.976 | 56.971 | **4.301** | **52.166** | **3.353** | **51.666** | **4.358** | 59.224 | **3.632** | 57.173 |
| WK-mr | 14.092 | 64.341 | **3.907** | **58.195** | **4.416** | **57.898** | **3.767** | **46.905** | **6.898** | **51.888** |

Online dynamics finetuning aims to eliminate distribution shift by learning a more accurate dynamics model. The total variation distance of the dynamics model from the true environment on a relatively uniform state-action distribution can reflect the extent of distribution shift. We calculate this distance on fake buffer before and after dynamics finetuning. The fake buffer is generateed by the offline dynamics and policy during offline training. The results are summarized in Table 7. From the Table 7, we observe that most of the online dynamics finetuning significantly reduces the distance $d_{\text{TV}}\left(P_{\widehat{\mathcal{M}}}, P_{\mathcal{M}}\right)$ between the real environment on the fake buffer.

# E AVERAGE IMPROVEMENT ACROSS ALL TASKS

In order to be more straightforward to compare the average improvement of our method DOOF with other baselines, we compute average improvement across tasks (HC, HP, WK) under different dataset qualities (r, m, mr) and task P under different datasets (c, h) in Table 8. As the results showed, our method outperform other baselines on Gym tasks. While FOWM shows the best performance on the Adroit tasks, our method shows better performance than other baselines except FOWM and PEX.

Table 8: The aggregation of random quality dataset of HC, HP and WK environment is denoted as R, similarly, the medium quality dataset aggregation is denoted as M and the medium-replay denoted as MR. The aggregation of pen-cloned and pen-human dataset is denoted as P. The average performance of our method DOOF and other baseline methods across tasks(HC, HP, WK) under aggregation of different dataset qualities(R, M, MR) and average performance of task P are computed. The gray results means this finetuning does not improve the performance and the **bold** results represent the best among these algorithms.

| | DOOF | IQL | | CaL-QL | | CQL | |
|---|---|---|---|---|---|---|---|
| | OFF-300K | ON-10K | OFF-300K | ON-10K | OFF-300K | ON-10K | OFF-300K |
| R | 24.5 **34.7**(+**10.2**) | 8.7 8.5(− 0.2) | 7.1(− 1.6) | 12.1 5.8(− 6.3) | 1.5(−10.6) | 10.4 6.3(− 4.1) | 4.1(− 6.3) |
| M | 72.2 **98.2**(+**26.0**) | 63.0 62.6(− 0.4) | 62.4(− 0.6) | 62.7 68.3(+ 5.6) | 68.6(+ 5.9) | 63.4 65.3(+ 1.9) | 63.7(+ 0.3) |
| MR | 73.8 **97.8**(+**24.0**) | 72.9 72.7(− 0.2) | 76.3(+ 3.4) | 74.6 77.3(+ 2.7) | 78.1(+ 3.5) | 70.3 74.1(+ 3.8) | 77.8(+ 7.5) |
| P | 43.8 58.1(+14.3) | 63.0 62.8(− 0.2) | 64.6(+ 1.6) | 2.1 −4.0(− 6.1) | −4.1(− 6.2) | −3.4 −3.7(− 0.3) | −4.6(− 1.2) |

| | DOOF | SPOT | | PEX | | FOWM | |
|---|---|---|---|---|---|---|---|
| | OFF-300K | ON-10K | OFF-300K | ON-10K | OFF-300K | ON-10K | OFF-300K |
| R | 24.5 **34.7**(+**10.2**) | 6.0 7.9(+ 1.9) | 14.4(+ 8.4) | 11.0 10.9(− 1.0) | 8.5(− 2.5) | 9.6 11.8(+ 2.2) | 11.8(+ 2.2) |
| M | 72.2 **98.2**(+**26.0**) | 61.6 62.0(+ 0.4) | 62.8(+ 1.2) | 59.9 41.9(−18.0) | 61.5(+ 1.6) | 31.0 50.0(+19.0) | 40.9(+9.9) |
| MR | 73.8 **97.8**(+**24.0**) | 66.4 57.3(− 9.1) | 70.2(+ 3.8) | 64.8 69.9(+ 5.1) | 59(− 5.8) | 58.8 64.2(+ 5.4) | 64.8(+ 6.0) |
| P | 43.8 58.1(+14.3) | 14.2 14.6(+ 0.4) | 20.0(+ 5.8) | 42.0 44.4(+ 2.4) | 58.7(+16.7) | −1.4 26.4(+27.8) | **32.8**(+**34.2**) |

## F  SOCIAL IMPACT

Our DOOF framework, by effectively decoupling policy optimization from distribution shift elimination, promises to enhance the reliability and efficiency of AI systems. This advancement not only facilitates safer deployment in critical domains such as healthcare and autonomous vehicles but also promotes equitable and sustainable development across various sectors by optimizing data usage and reducing computational costs.

