# OpenReview forum: "Decoupled Offline to Online finetuning via Dynamics Model"
_ICLR.cc/2025/Conference — Submitted to ICLR 2025_

### Official Review · Reviewer_JnJh · 2024-10-28

**Soundness:** 3
**Presentation:** 3
**Contribution:** 3
**Rating:** 6
**Confidence:** 4

**Summary:**

The authors study the problem of an offline to online fine-tuning framework in model-based RL. To address the trade-off between policy optimization and distribution shift elimination, DOOF (Decoupled Offline to Online Fine-tuning) suggests a method that decouples the elimination of distribution shift from the policy improvement process. This means they utilize the fine-tuned dynamics model trained with OOD actions and unbalanced states to assist in the fine-tuning of the policy in an offline mode, without additional interaction with the environment.

**Strengths:**

* I think the paper provides a clear intuition for offline to online RL using a model-based approach. In equation (5), the performance difference represents the expected dynamic distance of the offline dataset, conservatism on OOD actions, and unbalanced state distribution. Also, they show the distribution of total variation distance and uncertainty figures. It clearly explains what data is needed for fine-tuning the dynamics.
* Compared to existing offline RL algorithms, DOOF has high performance, despite only 10k online interaction steps. It significantly reduces the sample complexity.

**Weaknesses:**

* In Section 3.1, the authors assume minimal expected distance within the distribution of the offline dataset, overlooking a key limitation: even if specific state-action pairs are present in the offline dataset, the small sample size of some pairs can lead to significant uncertainty between the learned and true dynamics. I recommend that the authors address this limitation directly, discussing its implications for the reliability of their method in practice, particularly under conditions of sparsity in the offline dataset.
   * This issue is underscored in prior work on model-based offline RL, as noted in Theorem 1 of [1].
   * Figure 1 (the distribution of total variation distance on the WK-m dataset) further illustrates this, where the orange distribution (offline dataset) shows substantial overlap with the blue distribution (OOD actions).

* The theoretical results presented indeed appear to rely substantially on prior work by Yu et al. (2020) and Luo et al. (2018), without providing additional theoretical guarantees, such as a formal proof that the policy trained by the DOOF algorithm converges to the optimal policy. This raises questions about the potential advantages of DOOF over existing algorithms. To strengthen their theoretical contribution, the authors should explore convergence guarantees or derive performance bounds for their decoupled approach, highlighting how it compares with established methods in the field.

* The uncertainty u(s,a) indeed relies on a strong assumption, as seen in Yu et al. (2020). More recent works ([1] and [2]) have addressed this by redefining u(s,a) based on state-action pair frequency and Bellman estimation inconsistency, potentially offering a more precise measure. How about comparing their uncertainty measure with these newer approaches and discussing the benefits or limitations of incorporating these alternative interpretations within their framework?

[1] Kim, Byeongchan, and Min-hwan Oh. "Model-based offline reinforcement learning with count-based conservatism." International Conference on Machine Learning. PMLR, 2023.

[2] Sun, Yihao, et al. "Model-Bellman inconsistency for model-based offline reinforcement learning." International Conference on Machine Learning. PMLR, 2023.

**Questions:**

In addition to questions about potential weaknesses, I would like to raise a few more questions:

* Observing Equation (6), modifying the reward to encourage optimistic exploration appears very similar to UCB methods, which are commonly used in online RL.
* In Table 3, the distance differences seem almost identical regardless of the value of $\lambda_{on}$. Does this imply that simply collecting synthetic data with optimistic rewards suffices?
* $\lambda_{on}$ is also a hyperparameter; what is the optimal way to select it?
* In Appendix C.3, the offline penalty coefficient differs from the MOPO official coefficient. Did they tune it separately?

---

> ### Author Response · Authors · 2024-11-24
> **Response to Reviewer JnJh (1/2)**
>
> Thank you for the valuable comments. We provide clarification on your concerns as follows. We hope these new theoretical and experimental results can address your concerns, and we look forward to your further discussion.
>
> $~$
>
> **W1**: In Section 3.1, the authors assume minimal expected distance within the distribution of the offline dataset, overlooking a key limitation: even if specific state-action pairs are present in the offline dataset, the small sample size of some pairs can lead to significant uncertainty between the learned and true dynamics. I recommend that the authors address this limitation directly, discussing its implications for the reliability of their method in practice, particularly under conditions of sparsity in the offline dataset.
>
> - This issue is underscored in prior work on model-based offline RL, as noted in Theorem 1 of [1].
> - Figure 1 (the distribution of total variation distance on the WK-m dataset) further illustrates this, where the orange distribution (offline dataset) shows substantial overlap with the blue distribution (OOD actions).
>
> **A1:** In Line 202~203, we state that "the expected distance within the distribution of the offline dataset (the first term a)) should be minimal". What we intend to convey is that, the expectation of the distance on offline dataset $(s_{\mathcal{D}}, a_{\mathcal{D}})$ is largely smaller than the expected distance on OOD state-action pairs, $(s_{\mathcal{D}}, a_{\neg \mathcal{D}})$ or $(s_{\neg \mathcal{D}}, a)$. Besides, this statement can also be supported by the Theorem 1 of [1], which has been included in *Discussion of Theorem 3.1 and 3.2* in new paper version。
>
> The reviewer's concern is that the expected distance on the offline dataset $(s_{\mathcal{D}}, a_{\mathcal{D}})$ be larger due to the infrequent occurrence of certain $(s,a)$. This situation can indeed arise, as illustrated in Figure 1, but it does not conflict with our statement.
>
> $~$
>
> **W2:** The theoretical results presented indeed appear to rely substantially on prior work by Yu et al. (2020) and Luo et al. (2018), without providing additional theoretical guarantees, such as a formal proof that the policy trained by the DOOF algorithm converges to the optimal policy. This raises questions about the potential advantages of DOOF over existing algorithms. To strengthen their theoretical contribution, the authors should explore convergence guarantees or derive performance bounds for their decoupled approach, highlighting how it compares with established methods in the field.
>
> **A2:** On the one hand, we acknowledge that the theoretical aspect of this paper is indeed somewhat limited. With the assistance of reference [1], we have supplemented our discussion with guarantees regarding online dynamics finetuning and offline policy finetuning. We are grateful for the valuable feedback and crucial references provided by the reviewers. We have added the following theoretical results to verify the effectiveness of our proposed DOOF:
>
> - Online dynamics finetuning can indeed eliminate distribution shift by learning a more accurate dynamics model, refer to *Discussion of Theorems 3.1 and 3.2* for more details.
>
> - Offline policy finetuning can lead to performance improvements compared to the offline pre-trained policy $\pi_{\text{off}}$, refer to *Discussion of Theorems 3.3* for more details.
>
> On the other hand, our contribution lies in analyzing the offline-to-online problem from a decoupled perspective within the model-based framework. Although we are not the first to study the offline-to-online in a model-based context, we are the first to introduce a decoupled perspective, fully unleashing the potential of model-based RL in addressing this issue.
>
> $~$
>
> **W3**: The uncertainty u(s,a) indeed relies on a strong assumption, as seen in Yu et al. (2020). More recent works ([1] and [2]) have addressed this by redefining u(s,a) based on state-action pair frequency and Bellman estimation inconsistency, potentially offering a more precise measure. How about comparing their uncertainty measure with these newer approaches and discussing the benefits or limitations of incorporating these alternative interpretations within their framework?
>
> **A3:** Our theoretical framework is not built upon the specific form of $u(s,a)$, which makes the proposed decoupled framework universally applicable. This decoupled framework is not only suitable for MOPO but also for any other uncertainty-based methods, including Count-MORL [1] and MOBILE [2]. We have chosen the simplest and most general algorithm to validate the rationality of this framework.

---

> ### Author Response · Authors · 2024-11-24
> **Response to Reviewer JnJh (2/2)**
>
> **Q1**: Observing Equation (6), modifying the reward to encourage optimistic exploration appears very similar to UCB methods, which are commonly used in online RL.
>
> **A4:** In the paper [3], the conservative term in offline reinforcement learning (RL) is linked to the concept of anti-exploration. From this perspective, the conservative reward $\tilde{r} = r - u$ in MOPO can also be viewed as a form of anti-exploration. Furthermore, the uncertainty policy in DOOF, trained under the guidance of $r^u = r + u$, can be seen as encouraging exploration.
>
> [3] Rezaeifar S, Dadashi R, Vieillard N, et al. Offline reinforcement learning as anti-exploration[C]//Proceedings of the AAAI Conference on Artificial Intelligence. 2022, 36(7): 8106-8114.
>
> $~$
>
> **Q2**: In Table 3, the distance differences seem almost identical regardless of the value of λon. Does this imply that simply collecting synthetic data with optimistic rewards suffices?
>
> **A5:** In fact, different values of $\lambda_{on}$ in Table 3 lead to varying distances. Some values of $\lambda_{on}$ may paradoxically result in an increased distance after online dynamics finetuning.
>
> Merely interacting with the offline dynamics to collect synthetic data with optimistic rewards is insufficient. Optimistic rewards can cause the collected synthetic data to be out-of-distribution (OOD) compared to the offline dataset, which cannot ensure accuracy.
>
> $~$
>
> **Q3**: λon is also a hyperparameter; what is the optimal way to select it?
>
> **A6:** $\lambda_{\text{on}}$ is a hyperparameter that needs finetuning and is closely related to the penalty coefficient $\lambda_{\text{off}}$ of offline training. We use this ratio $\lambda_{\text{on}}:\lambda_{\text{off}} = ratio$ to indirectly choose the hyperparameter $\lambda_{\text{on}}$.
>
> $~$
>
> **Q4**: In Appendix C.3, the offline penalty coefficient differs from the MOPO official coefficient. Did they tune it separately?
>
> **A7:** we do not use the official implementation of MOPO. Instead, we use the offlineRL-kit repository(https://github.com/yihaosun1124/OfflineRL-Kit) which the implementation shows better performance than official MOPO repository. We also implement the experiments based on the offlineRL-kit, rather than MOPO official implementation.

---

> > ### Comment · Reviewer_JnJh · 2024-11-26
> >
> > I appreciate the authors' detailed response to my comments. However, I still have a few remaining concerns.
> >
> > - The authors state, "We propose a decoupled offline-to-online fine-tuning framework using the dynamics model from model-based methods." However, in the offline policy improvement phase, synthetic data is generated using a pessimistic reward function, while in the online phase, data is collected with a reward function that promotes exploration. Therefore, it is difficult to see this framework as truly "decoupled" in the offline-to-online fine-tuning process. Instead, it appears that the offline and online methods are each appropriately utilized within the framework.
> > - The ratio between $\lambda_{\text{off}}$ and $\lambda_{\text{on}}$ seems insufficient for evaluating the method's effectiveness. This is because both $\lambda_{\text{off}}$ and $\lambda_{\text{on}}$ are hyperparameters that are optimized independently. Typically, $\lambda_{\text{off}}$ would be tuned first to achieve optimal performance, followed by tuning $\lambda_{\text{on}}$. In such cases, the ratio merely reflects the result of this two-step optimization process and does not necessarily provide meaningful insights.
> >    - For instance, if the performance is similar when $\lambda_{\text{off}} = 1$ and $\lambda_{\text{off}} = 2$, can we conclusively say that the performance will also be similar when the ratio is fixed at 2, such that $\lambda_{\text{on}} = 2$ and $\lambda_{\text{on}} = 4$?

---

> > > ### Author Response · Authors · 2024-11-26
> > > **Further Reponse (1/2)**
> > >
> > > We are extremely grateful for the further discussions. We are pleased that our previous responses have addressed the reviewer's concerns, particularly on the theoretical aspects. In response to the reviewer's remaining concerns, here are our replies:
> > >
> > > $~$
> > >
> > > > The authors state, "We propose a decoupled offline-to-online fine-tuning framework using the dynamics model from model-based methods." However, in the offline policy improvement phase, synthetic data is generated using a pessimistic reward function, while in the online phase, data is collected with a reward function that promotes exploration. Therefore, it is difficult to see this framework as truly "decoupled" in the offline-to-online fine-tuning process. Instead, it appears that the offline and online methods are each appropriately utilized within the framework.
> > >
> > > **A1:** Offline to online finetuning need to improve the policy while overcoming the distributional shift simultaneously. Traditional offline-to-online algorithms address these two challenges concurrently during the online phase. In contrast, DOOF tackles distributional shift during the online phase before proceeding to offline policy improvement. The term 'decoupled' here signifies the separation of these two challenges, allowing for their sequential resolution.

---

> > > ### Author Response · Authors · 2024-11-26
> > > **Further Reponse (2/2)**
> > >
> > > > The ratio between 𝜆off and 𝜆on seems insufficient for evaluating the method's effectiveness. This is because both 𝜆off and 𝜆on are hyperparameters that are optimized independently. Typically, 𝜆off would be tuned first to achieve optimal performance, followed by tuning 𝜆on. In such cases, the ratio merely reflects the result of this two-step optimization process and does not necessarily provide meaningful insights.
> > > >
> > > > - For instance, if the performance is similar when 𝜆off=1 and 𝜆off=2, can we conclusively say that the performance will also be similar when the ratio is fixed at 2, such that 𝜆on=2 and 𝜆on=4?
> > >
> > > **A2:** The reviewer's question can be summarized as ***why the choice of $\lambda_{\text{on}}$ is tuned by the ratio between $\lambda_{\text{off}}$ and $\lambda_{\text{on}}$***.
> > >
> > > Firstly, the offline reward is defined as $r_{\text{off}} = r - \lambda_{\text{off}} \cdot u(s,a)$, while the online is $r_{\text{on}} = r + \lambda_{\text{on}} \cdot u(s,a)$. In offline, the term $u(s,a)$ represents the uncertainty of the dynamics model for specific state-action pairs. During the online phase, it is **precisely** the collection of these uncertain state-action pairs that aids in finetuning the dynamics model, thereby mitigating the distributional shift.
> > >
> > > In other words, a larger $\lambda_{\text{off}}$ indicates greater uncertainty in the dynamics model. Greater uncertainty suggests that these state-action pairs should be more actively collected, which implies that $\lambda_{\text{on}}$ should be larger. Considering this **positive correlation** between $\lambda_{\text{off}}$ and $\lambda_{\text{on}}$, we used $\lambda_{\text{off}}$ as a reference when tuning $\lambda_{\text{on}}$. Thus, we introduced this ratio $\lambda_{\text{off}}:\lambda_{\text{on}}=2$. Of course, when tuning $\lambda_{\text{on}}$, it is possible to search for this hyperparameter from scratch without considering $\lambda_{\text{off}}$. However, this approach might be relatively more challenging.
> > >
> > > In the above example, we indeed cannot claim the performance of $\lambda_{\text{on}}=2$ is similar to that of $\lambda_{\text{on}}=4$. The significance of introducing the ratio is to avoid blindness when tuning the parameter $\lambda_{\text{on}}$. For instance, In the following table from Table 5 in the appendix, on HC-r dataset, $\lambda_{\text{on}}=0.5$ is optimal with $\lambda_{\text{off}}=0.25$. Then, on the HC-m dataset (where $\lambda_{\text{off}}=0.5$), we should prioritize considering $\lambda_{\text{on}}=1.0$ rather than $\lambda_{\text{on}}=0.5$. In this case, the ratios for HC-r and HC-m are both $\lambda_{\text{off}}:\lambda_{\text{on}}=2$. Furthermore, we can see that while $\lambda_{\text{on}}$ varies widely, the ratios are concentrated around 2 or 0.5. This phenomenon also supports the rationality of introducing the ratio.
> > >
> > > | Datasets | $\lambda_{\text{on}}$ | Ratio ($\lambda_{\text{on}}:\lambda_{\text{off}}$) |
> > > | -------- | --------------------- | -------------------------------------------------- |
> > > | HC-r     | 0.5                   | 2                                                  |
> > > | HC-m     | 1.0                   | 2                                                  |
> > > | HC-mr    | 1.0                   | 2                                                  |
> > > | HP-r     | 10.0                  | 2                                                  |
> > > | HP-m     | 2.5                   | 0.5                                                |
> > > | HP-mr    | 5.0                   | 2                                                  |
> > > | WK-r     | 1.5                   | 2                                                  |
> > > | WK-m     | 0.125                 | 0.25                                               |
> > > | WK-mr    | 1.25                  | 0.5                                                |
> > > | P-c      | 1.25                  | 0.5                                                |
> > > | P-h      | 0.25                  | 0.5                                                |
> > >
> > > $~$
> > >
> > > Lastly, we are once again deeply grateful for your valuable comments. We hope that our above responses can address your queries. Should our response solve your concerns, we hope the rating can be elevated.

---

> > > ### Author Response · Authors · 2024-11-30
> > >
> > > As the discussion period deadline approaches, we eager to ascertain whether our responses have addressed your concerns. If your initial concerns have been largely resolved, we hope that this can be reflected in a higher rating. We sincerely appreciate the time and effort you have dedicated to our work!

---

> > > > ### Comment · Reviewer_JnJh · 2024-12-01
> > > >
> > > > Thank you for the detailed response and for addressing my earlier concerns. However, I still find several aspects of this work challenging to understand fully.
> > > >
> > > > Specifically, in the offline RL phase, Theorems 3.1 and 3.3 appear to be directly adapted from MOPO, while Theorem 3.2 seems to rely on Count-MORL. Moreover, the lack of policy performance guarantees when transitioning from offline to online settings leaves me uncertain about this work's theoretical contributions.
> > > >
> > > > The claimed novelty of introducing a "decoupled perspective" appears to be a natural outcome of transitioning to online RL, where the reward function shifts from pessimistic to optimistic. Furthermore, the integration of model-based RL algorithms for offline and online settings seems more like a synthesis of existing approaches rather than a fundamentally novel contribution.

---

> > > > > ### Author Response · Authors · 2024-12-01
> > > > >
> > > > > Thank you for your further replies and discussions. The remaining concerns are all related to our **decoupled** framework:
> > > > >
> > > > > | Stages                     | Challenges to solve  |
> > > > > | -------------------------- | -------------------- |
> > > > > | Online Dynamics Finetuning | Distributional Shift |
> > > > > | Offline Policy Finetuning  | Policy Improvement   |
> > > > >
> > > > > Offline to online finetuning needs to achieve the policy improvement disturbed by the distributional shift. DOOF decouples these challenges. During online stage, DOOF finetunes dynamics model to address the distributional shift. Then during offline stage, DOOF finetunes policy in offline manner using a more accurate and finetuned dynamics model to achieve the policy improvement.
> > > > > ___
> > > > > $~$
> > > > > ___
> > > > > > Specifically, in the offline RL phase, Theorems 3.1 and 3.3 appear to be directly adapted from MOPO, while Theorem 3.2 seems to rely on Count-MORL. Moreover, the lack of policy performance guarantees when transitioning from offline to online settings leaves me uncertain about this work's theoretical contributions.
> > > > >
> > > > > **A1:** While the theorems originate from offline model-based methods, their applicability to analyzing the offline to online problem lies in the decoupled framework:
> > > > >
> > > > > - Theorems 3.1 and 3.2 provide guarantees for online finetuning to mitigate distributional shift.
> > > > > - Theorem 3.3 itself offers a guarantee for offline policy performance. However, Theorem 3.3 can be extended to provide performance guarantees of DOOF, **because the policy improvement in DOOF is also the same as offline manner**, thanks to the decoupled framework.
> > > > >
> > > > > In summary, our theoretical contributions still relate to the decoupled framework. When and only when within this framework, we are able to leverage offline theories to analyze the offline to online problem.
> > > > >
> > > > >
> > > > > $~$
> > > > >
> > > > > > The claimed novelty of introducing a "decoupled perspective" appears to be a natural outcome of transitioning to online RL, where the reward function shifts from pessimistic to optimistic. Furthermore, the integration of model-based RL algorithms for offline and online settings seems more like a synthesis of existing approaches rather than a fundamentally novel contribution.
> > > > >
> > > > > **A2:** MOORe and FOWM are also model-based methods for offline to online fine tuning. However, neither of them decouples the policy finetuning and the dynamics model, nor do they propose an optimistic reward function. Thus, our novelty is **not** a "natural outcome".
> > > > >
> > > > > Furthermore, in model-free offline to online finetuning methods, there is also no decoupled framework or optimistic reward. In conclusion, we do not agree with the claim regarding the novelty of DOOF.
> > > > > ___
> > > > > $~$
> > > > > ___
> > > > > We hope that our above responses can address your confusion regarding the theory and novelty. Thank you for your valuable feedback on our paper!

---

> > > > > > ### Comment · Reviewer_JnJh · 2024-12-01
> > > > > >
> > > > > > Thank you for the detailed response! After reviewing the author's response and the revised paper, I better understand the "decoupled" framework.
> > > > > >
> > > > > > In P5, lines 266-267, I think that it is essential to significantly emphasize the critical role of a more exploratory and optimistic policy in actively collecting OOD data. This approach is fundamental to addressing and mitigating distributional shifts. Explicitly linking this point to the discussion of extrinsic rewards will further underline its importance and ensure the reader fully grasps its relevance. Finally, adding the Count-MORL paper applied to Theorem 3.2 in the Related Work section will substantially strengthen the overall impact of the paper.

---

> > > > > > > ### Author Response · Authors · 2024-12-01
> > > > > > >
> > > > > > > Thanks for your valuable suggestions on the paper's writing. As we are unable to upload a new version of the paper at this time, we will include the following modifications in the future version:
> > > > > > >
> > > > > > > 1. Emphasize the role of exploratory and optimistic policy around Lines 266-267, as recommended by the reviewer.
> > > > > > > 2. Add the citation of Count-MORL in the related work section. The current references at Lines 257 and 726 do not sufficiently highlight Count-MORL.
> > > > > > >
> > > > > > > In the initial comments, the mentioned Count-MORL has helped us complete the theoretical guarantees for DOOF's online dynamics finetuning. The improvements in the theoretical section of our paper are also thanks to your valuable comments.
> > > > > > >
> > > > > > > $~$
> > > > > > > ___
> > > > > > >
> > > > > > > $~$
> > > > > > >
> > > > > > > We appreciate the time and effort you have dedicated to our paper. Through multiple rounds of discussion, concerns have been continuously addressed, and the paper has been progressively refined. If the reviewer feels that the concerns have been largely resolved, we hope for a rating increase as a happy ending to this series of discussions.
> > > > > > >
> > > > > > > Regardless, we wish the reviewer all the best in both research and life!

---

> > > > > > > > ### Comment · Reviewer_JnJh · 2024-12-01
> > > > > > > >
> > > > > > > > Thanks for all your responses and the time you took to discuss! The authors' thorough answers to my many questions helped me understand the parts I had overlooked!
> > > > > > > >
> > > > > > > > Therefore, I raise the score to 6!

---

> > > > > > > > > ### Author Response · Authors · 2024-12-01
> > > > > > > > >
> > > > > > > > > Once again, we extend our sincere gratitude for your insightful comments! The discussion with you has been extremely valuable and pleasant. Wishing you all the best!

---

### Official Review · Reviewer_fU5C · 2024-11-03

**Soundness:** 2
**Presentation:** 3
**Contribution:** 2
**Rating:** 3
**Confidence:** 4

**Summary:**

This paper introduces a model-based offline RL method that decouples environment dynamics model fine-tuning from policy optimization.
It adopts an offline-online-offline manner that differs from previous work mainly in that it only fine-tunes dynamics model in the online stage,
and introduce an additional offline stage to fine-tune policy based on extended dataset and fine-tuned dynamics model.

**Strengths:**

This paper introduces a model-based offline RL method that decouples environment dynamics model fine-tuning from policy optimization, which is an intuitively appealing motivation.

**Weaknesses:**

1. The idea of the article is relatively intuitive, but its effectiveness lacks theoretical proof. Additionally, there may be an error in Equation (5), as each of the three terms on the right side is missing a constant coefficient.
2. There is no formal definition of the three types of state-action pairs: ID state-action pairs, ID state but OOD action pairs, and totally OOD state-action pairs.
3. Experiments were only conducted on a subset of the datasets in D4RL (Halfcheetah, Hopper, Walker2D, Pen). Firstly, the computational cost of conducting experiments in D4RL is not high, and there are other environments in Mujoco like Ant, as well as environments in Adroit like Hammer, Door, and Relocate. D4RL also includes Maze-like environments and kitchen-like environments, among others. Secondly, most datasets used in this paper were collected by RL agents, and the feature distribution of these datasets differs significantly from that of data collected by human experts. Therefore, it is unknown whether the method is applicable to human data.
4. DOOF is a model-based offline RL method, but most of the experimental results are compared with model-free offline RL methods, as shown in Table 4. (According to the introduction in Section 4, only FOWM is a model-based method.)
5. The rationality of the evaluation indicators needs to be questioned. For example, in Section 5.2, TV is calculated on the data in the fake buffer, which satisfies a special distribution. Whether the TV calculated on this distribution is representative is unknown. Similarly, the results in Table 3 can only prove that DOOF can reduce TV on the data distribution represented by the fake buffer.
6. The experimental scenarios are not consistent. Table 4 provides experimental results for the four environments Halfcheetah, Hopper, Walker2D, and Pen, while Table 3 does not include experimental results for Pen. Similarly, the experimental results in Fig.1-4 also only include results from some of the environments.

**Questions:**

1. Regarding Equation (3), by splitting the left side into the sum of the two terms on the right side, can it be demonstrated that the first term will decrease when fine-tuning $\hat{M}$? When optimizing $\pi_{off}$, will the second term decrease without causing the first term to increase?
2. Is it reasonable to calculate TV on the data in the fake buffer? Why not randomly sample states and actions in the environment and calculate TV?
3. If online fine-tuning of the world model is possible, why not use a random policy to sample data in the environment and learn the world model entirely based on this data; or only use reward based on uncertainty and no longer use the original reward from the environment?
4. Fig.2 presents experimental results obtained on the specific task of Walker2D. Whether the insights gained from this result are universal and sufficient to support the motivation for designing the algorithm is questionable.

---

> ### Author Response · Authors · 2024-11-24
> **Response to Reviewer fU5C (1/2)**
>
> Thank you for the valuable comments. We provide clarification on your concerns as follows. We hope these new theoretical and experimental results can address your concerns, and we look forward to your further discussion.
>
> $~$
>
> **W1:** The idea of the article is relatively intuitive, but its effectiveness lacks theoretical proof. Additionally, there may be an error in Equation (5), as each of the three terms on the right side is missing a constant coefficient.
>
> **A1:** The most significant innovation of our paper lies in successfully decoupling the offline-to-online transition within the model-based framework into two distinct challenges: a) the elimination of distributional shift and b) policy improvement. Subsequently, we leverage online dynamics fine-tuning to address the distributional shift. We have added the following theoretical results to verify the effectiveness of our proposed DOOF:
>
> - Online dynamics fine-tuning can indeed eliminate distribution shift by learning a more accurate dynamics model, refer to *Discussion of Theorems 3.1 and 3.2* for more details.
>
> - Offline policy finetuning can lead to performance improvements compared to the offline pre-trained policy $\pi_{\text{off}}$, refer to *Discussion of Theorems 3.3* for more details.
>
> The Equation (5) in original paper is "$=$", while it should be "$\leq$". Thanks again for your careful review. In new version of the paper, this issue has been addressed in Equation (6).
>
> $~$
>
> **W2:** There is no formal definition of the three types of state-action pairs: ID state-action pairs, ID state but OOD action pairs, and totally OOD state-action pairs.
>
> **A2:** We apologize for the omission of a formal definition, which we have now included in the new paper version. The definitions are as follows:
>
> 1. ID state-action pairs $(s_{\mathcal{D}}, a_{\mathcal{D}})$ indicate that both the state and action belong to the offline dataset $\mathcal{D}$, i.e., $s \in \mathcal{D}$ and $a \in \mathcal{D}$.
> 2. ID state but OOD action pairs $(s_{\mathcal{D}}, a_{\neg \mathcal{D}})$ indicate that the state belongs to the offline dataset $\mathcal{D}$ while the action does not, i.e., $s \in \mathcal{D}$ and $a \not\in \mathcal{D}$.
> 3. Totally OOD state-action pairs $(s_{\neg \mathcal{D}}, a)$ indicate that the state does not belong to the offline dataset $\mathcal{D}$, and the action is arbitrary, i.e., $s \not\in \mathcal{D}$ and $a \in \mathcal{A}$. In this case, regardless of whether the action belongs to the offline dataset, the distribution is already significantly distant from the offline dataset, hence the term totally OOD state-action pairs.
>
> $~$
>
> **W3:** Experiments were only conducted on a subset of the datasets in D4RL (Halfcheetah, Hopper, Walker2D, Pen). Firstly, the computational cost of conducting experiments in D4RL is not high, and there are other environments in Mujoco like Ant, as well as environments in Adroit like Hammer, Door, and Relocate. D4RL also includes Maze-like environments and kitchen-like environments, among others. Secondly, most datasets used in this paper were collected by RL agents, and the feature distribution of these datasets differs significantly from that of data collected by human experts. Therefore, it is unknown whether the method is applicable to human data.
>
> **A3:** Offline model-based algorithms have performed poorly in other datasets, such as AntMaze, Adroit, and Kitchen. This poor performance hinders the ability to further enhance the algorithms through fine-tuning, preventing us from validating the performance of DOOF across a more diverse set of datasets.
>
> Besides, Pen-human (P-h) is the dataset that contains expert demonstration from human.
>
> $~$
>
> **W4:** DOOF is a model-based offline RL method, but most of the experimental results are compared with model-free offline RL methods, as shown in Table 4. (According to the introduction in Section 4, only FOWM is a model-based method.)
>
> **A4:**  In the context of model-based RL, the offline to online finetuning algorithms include: MOORe, and FOWM [1].
>
> - MOORe: Model-based Offline-to-Online Reinforcement Learning. MOORe directly adopts the balanced buffer from another model-free offline to online finetuning algorithm [2]. MOORe finetunes both the dynamics model and the policy through online interaction. The code is not public.
> - FOWM: Finetuning Offline World Models in the Real World. FOWM is based on TDMPC2 [3, 4] and regularizes the planner at test-time by balancing estimated returns and epistemic model uncertainty. During online finetuning, FOWM finetunes the dynamics model and the policy simultaneously. The official code has been open-sourced.
>
> MOORe simply transplants model-free approaches into model-based scenarios without any specific design considerations. Therefore, we have not included it in our baselines, focusing solely on the comparison with FOWM.

---

> ### Author Response · Authors · 2024-11-24
> **Response to Reviewer fU5C (2/2)**
>
> **W5:** The rationality of the evaluation indicators needs to be questioned. For example, in Section 5.2, TV is calculated on the data in the fake buffer, which satisfies a special distribution. Whether the TV calculated on this distribution is representative is unknown. Similarly, the results in Table 3 can only prove that DOOF can reduce TV on the data distribution represented by the fake buffer.
>
> **Q2:** Is it reasonable to calculate TV on the data in the fake buffer? Why not randomly sample states and actions in the environment and calculate TV?
>
> **A5:** **W5** and **Q2** essentially are the same issue, hence we address them collectively here.
>
> The fake buffer's states are derived from the offline dataset $\mathcal{D}$, while actions are sampled from the policy. Although the policy is conservative, there is still a certain probability of selecting out-of-distribution (OOD) actions, making the fake buffer a combination of $(s_{\mathcal{D}}, a_{\mathcal{D}})$ and $(s_{\mathcal{D}}, a_{\neg \mathcal{D}})$. Such a fake buffer indeed has its limitations.
>
> Therefore, following your suggestion, we have assessed this distance within a uniformly sampled dataset. The results are presented in Table 3 of the revised manuscript.
>
> $~$
>
> **W6:** The experimental scenarios are not consistent. Table 4 provides experimental results for the four environments Halfcheetah, Hopper, Walker2D, and Pen, while Table 3 does not include experimental results for Pen. Similarly, the experimental results in Fig.1-4 also only include results from some of the environments.
>
> **A6:** Figure 1 and 2 are used to illustrate the motivation. Besides, we have added results on other datasets in Appendix D. The conclusion derived from these datasets is similar.The new Figure 3 has contained all the datasets in new paper version.
>
> $~$
>
> **Q1.1:** Regarding Equation (3), by splitting the left side into the sum of the two terms on the right side, can it be demonstrated that the first term will decrease when fine-tuning M^?
>
> **Q1.1:** When optimizing πoff, will the second term decrease without causing the first term to increase?
>
> **A7.1:** First and foremost, we have supplemented Theorem 3.2, which, in conjunction with the previous Theorem 3.1, allows us to ensure that the first term in equation (3) can be addressed through online fine-tuning of the dynamic model.
>
> **A7.2:** Strictly speaking, we are not certain whether the first term increases or decreases. However, we can demonstrate that the policy $\pi^*_{\text{off}}$, obtained by optimizing $\pi^*_{\text{off}}$, is closer to the optimal policy. We have supplemented Definition 3.3, which leads us to this conclusion.
>
> $~$
>
> **Q3.1:** If online fine-tuning of the world model is possible, why not use a random policy to sample data in the environment and learn the world model entirely based on this data;
>
> **Q3.2:** or only use reward based on uncertainty and no longer use the original reward from the environment?
>
> **A8.1:** It is unnecessary and wasteful to learn the dynamic model from scratch, discarding the well-learned offline dynamics. Correcting the errors in the offline dynamic model is a more efficient approach.
>
> **A8.2:** Discarding the original reward is not reasonable. The online policy should collect data from regions with high rewards and high uncertainty (i.e., areas where the dynamics are not well learned). Using only the uncertainty reward would result in collecting data that includes a large number of low-reward areas, which would be of little help for downstream learning.
>
> $~$
>
> **Q4:** Fig.2 presents experimental results obtained on the specific task of Walker2D. Whether the insights gained from this result are universal and sufficient to support the motivation for designing the algorithm is questionable.
>
> **A9:** We use this example to illustrate the exact phenomenon, while similar occurrences can be observed in other datasets as well. To this end, we have supplemented additional experimental results from other datasets in Appendix D.2.
>
> $~$
>
> **Reference**
>
> [1] Feng Y, Hansen N, Xiong Z, et al. Finetuning offline world models in the real world[J]. arXiv preprint arXiv:2310.16029, 2023.
>
> [2] Lee S, Seo Y, Lee K, et al. Offline-to-online reinforcement learning via balanced replay and pessimistic q-ensemble[C]//Conference on Robot Learning. PMLR, 2022: 1702-1712.
>
> [3] Hansen N, Wang X, Su H. Temporal difference learning for model predictive control[J]. arXiv preprint arXiv:2203.04955, 2022.
>
> [4] Hansen N, Su H, Wang X. Td-mpc2: Scalable, robust world models for continuous control[J]. arXiv preprint arXiv:2310.16828, 2023.

---

> ### Author Response · Authors · 2024-11-26
>
> As the author-reviewer discussion period will end soon, we will appreciate it if you could check our response to your review comments. We are always willing to discuss with you about any further concerns or questions. If our response resolves your concerns, we hope the rating of this paper can be elevated. Thanks you again for the time and effort you have invested in our paper!

---

> > ### Comment · Reviewer_fU5C · 2024-11-27
> >
> > I would like to thank the authors for their detailed response to my concerns. Regarding your answer, I still have some questions:
> >
> > Q5 (About W1-A1): (1) In Theorem 3.1, an assumption is made that the two MDPs have the same reward function. Does this assumption hold in your settings, and does your trained world model predict rewards? (2) In Appendix A.2, during the proof of Theorem 3.2, how is the expectation symbol eliminated from equation 13 to equation 14?
> >
> > Q6 (About W2-A2): Could you please explain what the criteria are for determining "in-distribution"?
> >
> > Q7 (About W3-A3): (1) I believe that "Offline model-based algorithms have performed poorly in other datasets, such as AntMaze, Adroit, and Kitchen." cannot be a reason for selecting experimental tasks. Furthermore, could you briefly explain why offline model-based algorithms perform poorly on these tasks, and whether these reasons are limitations of offline model-based algorithms? (2) The Pen task belongs to Adroit; why didn't you conduct experiments on the other three tasks in Adroit, Hammer, Door, and Relocate? (3) Additionally, the only dataset that includes human demonstrations, Pen-Human, is a very special dataset. This dataset only includes 5k transitions (25 trajectories), while other datasets include at least 500k transitions. In your experiments, you sampled 10k online transitions, which is much more than Pen-Human but much less than other datasets. Therefore, I believe that the setting of the only experiment scenario that includes human demonstrations is very inconsistent with the settings of other scenarios. Hence, I think it is unknown whether the method is applicable to human data.
> >
> > Q8 (W4-A4): If, up to this point, there are only MOORe and FOWN among Model-based offline-to-online algorithms, and FOWN is significantly better than MOORe, I believe that for the comparative experimental section, it would be sufficient to only compare DOOF with FOWN. Comparing with other algorithms, such as Model-free Offline RL, I think it lacks practical significance because even the experimental settings are different.
> >
> > Q9 (W5,Q2-A5): The TV values calculated based on a uniformly sampled dataset are inherently high and show no significant decrease. Could you explain this?
> >
> > Q10 (W6-A6): Why are the results for the pen not provided in the analysis of the total variation distance (Table 2 and Appendix D)?
> >
> > Q11 (Q3.1-A8.1): (1) Continuously fine-tuning the world model turns into a continual learning problem. There are classic issues in continual learning, such as catastrophic forgetting and model plasticity. I believe that if a theoretical analysis cannot be provided to explain why fine-tuning is better, at least a set of experiments is needed to prove that fine-tuning is better than learning from scratch. (2) Additionally, my original question was whether, if online interaction is available, a world model trained on randomly sampled data would be more accurate than a world model trained on $\mathcal{D} \cup \mathcal{D}_{on}$. (Or to say, given the ability to sample online, why would it be better for the world model to predict more accurately in high-reward areas than to predict accurately across the entire state-action space with a uniform distribution?)

---

> > > ### Author Response · Authors · 2024-11-27
> > > **Further Reponse (1/2)**
> > >
> > > Thank you very much for engaging in further discussion with us! In response to the reviewer's remaining concerns, here are our replies:
> > >
> > > $~$
> > >
> > > > Q5 (About W1-A1): (1) In Theorem 3.1, an assumption is made that the two MDPs have the same reward function. Does this assumption hold in your settings, and does your trained world model predict rewards? (2) In Appendix A.2, during the proof of Theorem 3.2, how is the expectation symbol eliminated from equation 13 to equation 14?
> > >
> > > **A1:** (1) MOPO [1], Count-MORL [2], and MOBILE [3] all assume in their theoretical proofs that the reward function is known, hence 'the same reward function.' Such an assumption simplifies and clarifies the proofs. Moreover, these proofs are generalizable to scenarios where the reward function is unknown and must be learned [1]. In practice, world models (also known as dynamics models) all predict the reward function in these papers. Our DOOF also adopts the same setting as before.
> > >
> > > (2) The proof of Theorem 3.2 follows [2]. We have added a new formula in Appendix A.2 to provide a more detailed explanation of the transition from Equation 13 to 14. This formula utilizes the "subset" to scale from Equation 13 to 14. Due to the limitations of Markdown in displaying complex formulas, we recommend that the reviewer refer to the latest version of the paper for a clear representation.
> > >
> > > $~$
> > >
> > > > Q6 (About W2-A2): Could you please explain what the criteria are for determining "in-distribution"?
> > >
> > > **A2:** The state-action pairs sample from the offline dataset $\mathcal{D}$ can be viewed as the in-distribution data of $\mathcal{D}$. The distribution of $\mathcal{D}$ can be estimated by the VAE. Both BCQ [6] and SPOT [7] adopts the VAE to estimate the distribution of $\mathcal{D}$ and then use this distribution to determine whether one state-action pair is **in the distribution** of $\mathcal{D}$.
> > >
> > > $~$
> > >
> > > > Q7 (About W3-A3): (1) I believe that "Offline model-based algorithms have performed poorly in other datasets, such as AntMaze, Adroit, and Kitchen." cannot be a reason for selecting experimental tasks. Furthermore, could you briefly explain why offline model-based algorithms perform poorly on these tasks, and whether these reasons are limitations of offline model-based algorithms? (2) The Pen task belongs to Adroit; why didn't you conduct experiments on the other three tasks in Adroit, Hammer, Door, and Relocate? (3) Additionally, the only dataset that includes human demonstrations, Pen-Human, is a very special dataset. This dataset only includes 5k transitions (25 trajectories), while other datasets include at least 500k transitions. In your experiments, you sampled 10k online transitions, which is much more than Pen-Human but much less than other datasets. Therefore, I believe that the setting of the only experiment scenario that includes human demonstrations is very inconsistent with the settings of other scenarios. Hence, I think it is unknown whether the method is applicable to human data.
> > >
> > > **A3:** (1) Only when the offline policy has a certain level of performance does online finetuning become meaningful. Poor offline performance results in no subsequent performance improvements, thereby failing to validate the effectiveness.
> > >
> > > We cannot provide an exact reason why offline model-based algorithms underperform compared to model-free in these environments, but our literature review reveals that few uncertainty-based offline algorithms have reported results on these settings [1] [2] [3]. This limitation hinders DOOF from offering a more diverse set of experiments, as DOOF is based on the  uncertainty-based model-based offline algorithms.
> > >
> > > (2) Only MOBILE [3] reports the results on Adroit (but no relocate), and the performance is nearly zero expect for Adroit. we only choose pen for experiments.
> > >
> > > | datasets      | MOPO | MOBILE |
> > > | ------------- | ---- | ------ |
> > > | pen-human     | 10.7 | 30.1   |
> > > | door-human    | -0.2 | -0.2   |
> > > | hammer-human  | 0.3  | 0.4    |
> > > | pen-cloned    | 54.6 | 69.0   |
> > > | door-cloned   | 15.3 | 24.0   |
> > > | hammer-cloned | 0.5  | 1.5    |
> > >
> > > (3) In other works about offline to online finetuning [4] [5], there has been no special experimental setup due to the limited amount of data on pen-human. Following this conventions of the offline RL community, the 10k setting is also acceptable.
> > >
> > > $~$
> > >
> > > > Q8 (W4-A4): If, up to this point, there are only MOORe and FOWN among Model-based offline-to-online algorithms, and FOWN is significantly better than MOORe, I believe that for the comparative experimental section, it would be sufficient to only compare DOOF with FOWN. Comparing with other algorithms, such as Model-free Offline RL, I think it lacks practical significance because even the experimental settings are different.
> > >
> > > **A4:** Given that the existing algorithms for offline to online fine-tuning are predominantly focused on the model-free domain, we have also included additional model-free algorithms in our baseline comparisons

---

> > > ### Author Response · Authors · 2024-11-27
> > > **Further Reponse (2/2)**
> > >
> > > > Q9 (W5,Q2-A5): The TV values calculated based on a uniformly sampled dataset are inherently high and show no significant decrease. Could you explain this?
> > >
> > > **A5:** The entire state-action space is significantly larger compared to the offline dataset. The dynamics model can only ensure that it is close to the true dynamics on the region that is near the offline dataset, meaning a smaller Total Variation (TV) distance in those regions. In other areas, the dynamics model has hardly seen any data, so the TV distance is large.
> > >
> > > Furthermore, the reason for the less pronounced decrease is also due to the enormous size of the state-action space. A policy trained with $r+u$ cannot cover the entire state-action space, and therefore, it cannot achieve a significant reduction in TV distance across the entire space.
> > >
> > > $~$
> > >
> > > >Q10 (W6-A6): Why are the results for the pen not provided in the analysis of the total variation distance (Table 2 and Appendix D)?
> > >
> > > **A6:** In order to compute $d_{\text{TV}}$ on a certain dataset $(s_t,a_t)$, we need to set the true environment with state $s_t$ using the following code:
> > >
> > > ```python
> > > def _set_state_from_obs(self, obs:np.ndarray) -> None:
> > > 	if len(obs) == (self.env.model.nq + self.env.model.nv - 1):
> > > 		xpos = np.zeros(1)
> > > 		obs = np.concatenate([xpos, obs])
> > > ```
> > >
> > > For walker2d, hopper and halfcheetah,  *len(obs) == (self.env.model.nq + self.env.model.nv - 1)* is true while pen is False. As a result, the subsequent code will report an error on dataset pen.
> > >
> > > $~$
> > >
> > > > Q11 (Q3.1-A8.1): (1) Continuously fine-tuning the world model turns into a continual learning problem. There are classic issues in continual learning, such as catastrophic forgetting and model plasticity. I believe that if a theoretical analysis cannot be provided to explain why fine-tuning is better, at least a set of experiments is needed to prove that fine-tuning is better than learning from scratch. (2) Additionally, my original question was whether, if online interaction is available, a world model trained on randomly sampled data would be more accurate than a world model trained on 𝐷∪𝐷𝑜𝑛. (Or to say, given the ability to sample online, why would it be better for the world model to predict more accurately in high-reward areas than to predict accurately across the entire state-action space with a uniform distribution?)
> > >
> > > **A7:** The key of both these issues lies in the volume of data .
> > >
> > > (1) If there were no restrictions on the amount of data, or the number of online interactions, learning a dynamics model from scratch would undoubtedly outperform a finetuned dynamics model. However, in the context of offline-to-online fine-tuning and offline RL, we typically assume that data is costly and interactions are limited. As a result, training a dynamics model from scratch with only 10k data would certainly not be as effective as finetuning an offline pretrained dynamics model.
> > >
> > > (2) A more accurate dynamics model across the entire state-action space would certainly be preferable, but this requires a substantial amount of data. With limited data, we can only learn about limited region. Regions with higher returns are more likely to lead to better policies, hence the policy needs to maximize return. Concurrently, areas where the offline dynamics model is less accurate should be the focus of learning more, thus the policy also needs to maximize the uncertainty $u$. So we use the policy trained by $r+u$ rather than a random policy.
> > >
> > > $~$
> > >
> > > **Reference**
> > >
> > > [1] Yu T, Thomas G, Yu L, et al. Mopo: Model-based offline policy optimization[J]. Advances in Neural Information Processing Systems, 2020, 33: 14129-14142.
> > >
> > > [2] Kim, Byeongchan, and Min-hwan Oh. "Model-based offline reinforcement learning with count-based conservatism." International Conference on Machine Learning. PMLR, 2023.
> > >
> > > [3] Sun, Yihao, et al. "Model-Bellman inconsistency for model-based offline reinforcement learning." International Conference on Machine Learning. PMLR, 2023.
> > >
> > > [4] Li J, Hu X, Xu H, et al. Proto: Iterative policy regularized offline-to-online reinforcement learning[J]. arXiv preprint arXiv:2305.15669, 2023.
> > >
> > > [5] Lei K, He Z, Lu C, et al. Uni-o4: Unifying online and offline deep reinforcement learning with multi-step on-policy optimization[J]. arXiv preprint arXiv:2311.03351, 2023.\
> > >
> > > [6] Fujimoto S, Meger D, Precup D. Off-policy deep reinforcement learning without exploration[C]//International conference on machine learning. PMLR, 2019: 2052-2062.
> > >
> > > [7] Wu J, Wu H, Qiu Z, et al. Supported policy optimization for offline reinforcement learning[J]. Advances in Neural Information Processing Systems, 2022, 35: 31278-31291.

---

> > > ### Author Response · Authors · 2024-11-30
> > >
> > > As the discussion period deadline approaches, we eager to ascertain whether our responses have addressed your concerns. If your initial concerns have been largely resolved, we hope that this can be reflected in a higher rating. We sincerely appreciate the time and effort you have dedicated to our work!

---

> > > > ### Comment · Reviewer_fU5C · 2024-11-30
> > > >
> > > > I would like to thank the authors for their detailed response to my concerns. I still have some questions:
> > > >
> > > > Q12 (Q5(2)): The derivation of Equation (14) appears to be problematic; the $\ge$ sign should only hold for the specific $(s,a)$ that minimizes $d_{TV}(\cdot,\cdot)$.
> > > >
> > > > Q13 (Q5(1) and Q7): I believe that following the convenience of previous research is not always correct. (1) If the premises of the formula cannot be met in your experiments, then where lies the significance of simplifying the derivation process? (2) One cannot only select experimental scenarios that are advantageous to himself. If I am a follower of DOOF, I would prefer to know what the limitations of DOOF are, where the boundaries are, and whether it can be applied to my experimental scenarios.

---

> > > > > ### Author Response · Authors · 2024-11-30
> > > > >
> > > > > We are immensely grateful for the authors' further discussion! We are delighted that our previous responses have alleviated the majority of the reviewers' concerns. Below are our responses to the reviewers' additional concerns:
> > > > > ___
> > > > > ___
> > > > > > Q12 (Q5(2)): The derivation of Equation (14) appears to be problematic; the $\geq$ sign should only hold for the specific (s,a) that minimizes $d_{\text{TV}}$.
> > > > >
> > > > > **A1:** Actually, the derivation of Equation (14) is **correct** and does not rely on the specific $(s,a)$ that should minimizes $d_{\text{TV}}(\cdot, \cdot)$. The $\geq$ holds because $\sum[P(s_i,a_i) \cdot d_{\text{TV}}(s_i,a_i)] \geq P(s,a) \cdot d_{\text{TV}}(s,a)$ where $s,a \in (s_i,a_i)$ is one specific datapoint from the whole distribution.
> > > > >
> > > > > $~$
> > > > > > Q13 (Q5(1) and Q7): I believe that following the convenience of previous research is not always correct. (1) If the premises of the formula cannot be met in your experiments, then where lies the significance of simplifying the derivation process? (2) One cannot only select experimental scenarios that are advantageous to himself. If I am a follower of DOOF, I would prefer to know what the limitations of DOOF are, where the boundaries are, and whether it can be applied to my experimental scenarios.
> > > > >
> > > > > **A2.1:** DOOF can be viewed as an offline to online finetuning method based on MOPO, so the theoretical assmuptation and experiment setting are the **same** as MOPO. And the simplification of derivation process also holds.
> > > > >
> > > > > **A2.2:** Very appreciate to this valuable suggestions! We agree that it is essential to include a section on limitations to clarify under what circumstances DOOF may not be effective.
> > > > >
> > > > > Specifically, the boundaries of DOOF align with those of existing offline model-based methods. DOOF is effective in the domains where offline model-based methods are effective. In other words, the capabilities of offline model-based methods set the limits on the applicability of DOOF.
> > > > >
> > > > > $~$
> > > > >
> > > > > **The future modification** will contain 1) a more detailed description about Equation 14 and 2) an additional section about the limitation of DOOF in appendix.
> > > > >
> > > > > ___
> > > > > ___
> > > > >
> > > > > We would like to express our gratitude once again to the reviewer for their thorough comments. The time and effort you have dedicated will undoubtedly enhance the quality of our paper. We hope that our responses above can address your concerns.

---

> > > > > ### Author Response · Authors · 2024-12-01
> > > > >
> > > > > First and foremost, please allow me to express my sincere gratitude for the time and effort you have dedicated to this paper. Your comments have thoroughly assessed every aspect, including theoretical details and experimental setups. With your assistance, we have made the following improvements:
> > > > >
> > > > > 1. We have refined the theoretical aspects and their proof processes.
> > > > > 2. We have added the missing experimental results in Figure 3.
> > > > > 3. We have changed the 'fake buffer' into a more representative dataset which is uniformly sampled from the whole state-action space in Figure 2.
> > > > > 4. We have supplemented the results of Figures 1 and 2 on other datasets in Appendix D.
> > > > >
> > > > > Furthermore, we plan to include a section about limitations in future versions to elucidate the scope and reasons for the applicability of DOOF. This is an important lesson learned through the ongoing discussions with you.
> > > > >
> > > > > $~$
> > > > > ___
> > > > > $~$
> > > > >
> > > > > If your concerns and questions have been largely addressed, we hope that you can consider raising the score as a fitting ending to this meaningful discussion.
> > > > >
> > > > > Regardless, we are immensely grateful for your meticulous review of our paper and the valuable feedback provided. Wish you success in your research and happiness in your life!

---

> ### Comment · Reviewer_fU5C · 2024-12-02
>
> I would like to express my gratitude to the authors for their elaborate response to my concerns. During the rebuttal period, there have been changes to the manuscript. The formulaic expressions have become more lucid, and a portion of the experiments has been augmented. Nevertheless, your response only partially addresses my concerns: First, the formula reasoning needs to be further improved. For example, I believe that the first line of Equation (14) should be $\int\left[P(s_t,a_t)d_{\mathrm{TV}}\left(P_{\widehat{\mathcal{M}}}\left(s_t,a_t\right),P_{\mathcal{M}}\left(s_t,a_t\right)\right)^2\right]\mathrm{d}(s_t,a_t)$. The deduction process from Eq.(13) to Eq.(14) is problematic. The assumption of "the same reward function" in Theorem 3.1 does not match the actual implementation of your algorithm, and this gap between theory and practice can severely limit the effectiveness of the algorithm. Second, the experimental scenarios are relatively few and simple, and the existence of coding bugs in the environment should not be a reason for the lack of relevant experimental results. In addition, more appropriate comparison algorithms and evaluation indicators should be selected based on the motivation. I hope you can get some inspiration from our discussion. I am inclined to maintain my score.

---

> > ### Author Response · Authors · 2024-12-02
> >
> > Thank you for your reply! We regret that we have not been able to fully address all of your concerns. However, regarding your final claim, we believe there are some corrections that need to be made:
> >
> > > For example, I believe that the first line of Equation (14) should be $\int\left[P(s_t,a_t)d_{\mathrm{TV}}\left(P_{\widehat{\mathcal{M}}}\left(s_t,a_t\right),P_{\mathcal{M}}\left(s_t,a_t\right)\right)^2\right]\mathrm{d}(s_t,a_t)$. The deduction process from Eq.(13) to Eq.(14) is problematic.
> >
> > A1. There was indeed a typographical error in Equation (14), where $(s_i,a_i)$ should be $(s_t,a_t)$. We appreciate the reviewer for pointing out the issue. But the proof of Theorem 3.2 is **correct**, which can be verified by referring to the original appendix of Count-MORL.
> >
> > > The assumption of "the same reward function" in Theorem 3.1 does not match the actual implementation of your algorithm, and this gap between theory and practice can severely limit the effectiveness of the algorithm.
> >
> > A2. In MOPO, the third footnote in page 7 states "*If the reward function is known, we do not have to estimate the reward. The theory in Sections 4.1 and 4.2 applies to the case where the reward function is known. To extend the theory to an unknown reward function, we can consider the reward as being concatenated onto the state*". In theory, MOPO assumes that the reward function is known，also "the same reward function", and then generalizes to unknown scenarios during implementation. The offline algorithm for DOOF is MOPO, hence there is **no gap** in DOOF.
> >
> > > Second, the experimental scenarios are relatively few and simple,……
> >
> > A3. Offline to online algorithms (such as DOOF) are based on offline methods (such as MOPO). Therefore, the applicability of DOOF is unavoidably limited by MOPO.
> >
> > $~$
> > ___
> > $~$
> >
> > Thank you again for your thorough review of our paper. We wish you all the best!

---

### Official Review · Reviewer_Moig · 2024-11-04

**Soundness:** 3
**Presentation:** 3
**Contribution:** 2
**Rating:** 6
**Confidence:** 5

**Summary:**

The paper aims to address the decoupling of distribution shifts and policy improvement. They propose to only fine-tune the dynamics model online to address the distribution shift while leaving the policy improvement stage still performed on offline data. They employ empirical studies on classical offline RL datasets and demonstrate the effectiveness within the offline to online problems.

**Strengths:**

+ The paper provides a great summary of different state-of-the-art algorithms in Table 1, showing clearly how different algorithms approach offline to online fine-tuning problems;

+ The paper analyzes the distributions and shows them in clear visualization to demonstrate for example the reason for using an additional reward term.

**Weaknesses:**

The algorithmic innovations seem limited in the paper, it looks like moving the policy fine-tuning from the online stage into the offline stage with the online collected datasets used for fine-tuning the dynamics model.

**Questions:**

Since we have an online stage to finetune the dynamics model, what are the actual advantages of using offline methods to fine-tune the policy instead of using the online methods?

---

> ### Author Response · Authors · 2024-11-24
> **Response to Reviewer Moig (1/1)**
>
> Thank you for the valuable comments. We provide clarification on your concerns as follows. We hope these new theoretical and experimental results can address your concerns, and we look forward to your further discussion.
>
> $~$
>
> **W1：** The algorithmic innovations seem limited in the paper, it looks like moving the policy fine-tuning from the online stage into the offline stage with the online collected datasets used for fine-tuning the dynamics model.
>
> **A1:** From the implementation perspective, our decoupled framework is indeed straightforward, as the reviewer has noted. Our **innovation** lies in being the first to decouple the offline to online problem into two distinct phases: online distributional shift elimination followed by offline policy performance improvement. Additionally, we have theoretically analyzed this framework, including:
>
> - Online dynamics finetuning can indeed eliminate distributional shift by learning a more accurate dynamics model, refer to *Discussion of Theorems 3.1 and 3.2* for more details;
> - Offline policy finetuning can lead to performance improvements compared to the offline pre-trained policy $\pi_{\text{off}}$, refer to *Discussion of Theorems 3.3* for more details.
>
> In our experiments, we further demonstrate that this innovative framework offers exceptionally high interaction efficiency.
>
> $~$
>
> **Q1:** Since we have an online stage to finetune the dynamics model, what are the actual advantages of using offline methods to fine-tune the policy instead of using the online methods?
>
> **A2:** Transitioning from offline algorithms to online algorithms often leads to a sudden drop in performance, as depicted in Figure 1 of Cal-ql [1]; this is not a desirable approach. Utilizing offline algorithms ensures consistency and avoids such a sudden performance decline.
>
> Furthermore, we have supplemented our paper with Theorem 3.3. leveraging it, we demonstrate that the policy obtained from offline finetuning indeed shows an improvement over the offline pretrained policy $\pi_{\text{off}}$.
>
> [1] Nakamoto M, Zhai S, Singh A, et al. Cal-ql: Calibrated offline rl pre-training for efficient online fine-tuning[J]. Advances in Neural Information Processing Systems, 2024, 36.

---

> ### Author Response · Authors · 2024-11-26
>
> As the author-reviewer discussion period will end soon, we will appreciate it if you could check our response to your review comments. We are always willing to discuss with you about any further concerns or questions. If our response resolves your concerns, we hope the rating of this paper can be elevated. Thanks you again for the time and effort you have invested in our paper!

---

> ### Author Response · Authors · 2024-12-01
>
> The primary concern (also the weakness), mainly consists in the *novelty* of our proposed DOOF. Through discussions with other reviewers, we have formulated a better response:
>
> The *novelty* of DOOF lies in the **decoupled** framework for offline to online finetuning.
>
> | Stages                     | Challenges to solve  |
> | -------------------------- | -------------------- |
> | Online Dynamics Finetuning | Distributional Shift |
> | Offline Policy Finetuning  | Policy Improvement   |
>
> Offline to online finetuning needs to achieve the policy improvement disturbed by the distributional shift. DOOF decouples these challenges. During online stage, DOOF finetunes dynamics model to address the distributional shift. Then during offline stage, DOOF finetunes policy in offline manner using a more accurate and finetuned dynamics model to achieve the policy improvement.
>
> decoupling offers two significant advantages:
>
> - Theoretical aspect: The decoupled framework allows us to leverage existing offline model-based RL theory to analyze and address the offline to online finetuning. Specifically, although Theorems 3.1, 3.2, and 3.3 originate from MOPO [1] or Count-MORL [2], we can derive the guarantees for both online dynamics finetuning and offline policy finetuning. The key here lies in the **decoupled** framework.
> - Experimental aspect: This decoupled framework, equipped with an optimistic uncertainty policy, boasts exceptional interactive efficiency. The key is that the online phase focuses on eliminating distributional shift, laying the groundwork for policy improvement in the offline phase.
>
> [1] Yu T, Thomas G, Yu L, et al. Mopo: Model-based offline policy optimization[J]. Advances in Neural Information Processing Systems, 2020, 33: 14129-14142.
>
> [2] Kim, Byeongchan, and Min-hwan Oh. "Model-based offline reinforcement learning with count-based conservatism." International Conference on Machine Learning. PMLR, 2023.
>
> $~$
> ___
> $~$
>
> We hope that our above responses can address your concerns. We are always ready for further discussion with you. Should our responses successfully resolve your concerns, we hope you will consider increasing the score accordingly.
>
> Thank you once again for your efforts, and we wish you all the best.

---

> ### Author Response · Authors · 2024-12-02
>
> As the author-reviewer discussion phase is coming to an end, we are eagerly awaiting your feedback on whether your previous concerns have been addressed and if there are any further questions. If all your concerns have been resolved, we hope that you can consider raising the rating accordingly.
>
> Regardless, we wish you all the best!

---

> > ### Comment · Reviewer_Moig · 2024-12-03
> >
> > The reviewers have addressed my concerns. I'm happy to raise the score.

---

> > > ### Author Response · Authors · 2024-12-03
> > >
> > > Thank you for your affirmation of our work. We are delighted that we have addressed your concerns. This increase in rating serves as a perfect ending to this author-reviewer discussion phase. We wish you a pleasant life and success in your research!

---

### Official Review · Reviewer_pumy · 2024-11-04

**Soundness:** 3
**Presentation:** 3
**Contribution:** 3
**Rating:** 6
**Confidence:** 4

**Summary:**

The authors propose a decoupled model-based offline to online finetuning framework called DOOF. In the Online Finetuning stage, DOOF only updates the Dynamic model to mitigate the distribution shift, and then obtains Policy Improvement through Offline Finetuning. Experiments on D4RL validate the effectiveness of the method.

**Strengths:**

1. The authors propose an interesting perspective to solve the distribution shift problem in the Offline-to-Online stage, i.e., decoupling the elimination of distribution shift and policy improvement.
2. The authors provide a clear analysis of the sources of distribution error in the Offline-to-Online stage, which naturally leads to the proposed method.
3. The paper is clearly written.

**Weaknesses:**

The experimental design is not comprehensive enough.

**Questions:**

1. Why can't you compare with some specialized Offline-to-Online algorithms? You can set their Online interaction steps to 10k like yours, and report their Finetuning performance. If such a comparison is possible, I suggest increasing the Online interaction step budget to compare the final asymptotic performance.
2. The number of environments involved in the paper is limited. D4RL is a basic Offline Benchmark, and I suggest evaluating on more updated tasks.
3. Regarding the off(300k) baseline, does it mean that the initial Offline data, plus the 10k data from online interaction, are used for the final 300k offline gradient steps?

I'm willing to reconsider my score based on the new information provided in the Rebuttal stage.

---

> ### Author Response · Authors · 2024-11-24
> **Response to Reviewer pumy (1/1)**
>
> Thank you for the valuable comments. We provide clarification on your concerns as follows. We hope these new theoretical and experimental results can address your concerns, and we look forward to your further discussion.
>
> $~$
>
> **W1:** The experimental design is not comprehensive enough.
>
> **Q2**: The number of environments involved in the paper is limited. D4RL is a basic Offline Benchmark, and I suggest evaluating on more updated tasks.
>
> **A1:** **W1** and **Q2** essentially are the same issue, hence we address them collectively here.
>
> 1.For model-based offline to online baselines, only two algorithms are available:
>
> - MOORe: Model-based Offline-to-Online Reinforcement Learning. MOORe directly adopts the balanced buffer from another model-free offline to online finetuning algorithm [1]. MOORe finetunes both the dynamics model and the policy through online interaction. The code is not public.
> - FOWM: Finetuning Offline World Models in the Real World. FOWM is based on TDMPC2 [2, 3] and regularizes the planner at test-time by balancing estimated returns and epistemic model uncertainty. During online finetuning, FOWM finetunes the dynamics model and the policy simultaneously. The official code has been open-sourced.
>
> MOORe simply transplants model-free approaches into model-based scenarios without any specific design considerations. Therefore, we have not included it in our baselines, focusing solely on the comparison with FOWM.
>
> 2.For datasets, uncertainty-based offline model-based algorithms are only effective in Gym environments and Adroit's pen tasks. Their performance in other environments is extremely poor, which hinders further finetuning. As a result, the datasets that can be used to validate the DOOF are relatively limited.
>
> [1] Lee S, Seo Y, Lee K, et al. Offline-to-online reinforcement learning via balanced replay and pessimistic q-ensemble[C]//Conference on Robot Learning. PMLR, 2022: 1702-1712.
>
> [2] Hansen N, Wang X, Su H. Temporal difference learning for model predictive control[J]. arXiv preprint arXiv:2203.04955, 2022.
>
> [3] Hansen N, Su H, Wang X. Td-mpc2: Scalable, robust world models for continuous control[J]. arXiv preprint arXiv:2310.16828, 2023.
>
> $~$
>
> **Q1:** Why can't you compare with some specialized Offline-to-Online algorithms? You can set their Online interaction steps to 10k like yours, and report their Finetuning performance. If such a comparison is possible, I suggest increasing the Online interaction step budget to compare the final asymptotic performance.
>
> **A2:** The results of on(10k) are the finetuning results using only 10k online data. Since online interaction may be expensive or dangerous, we focus on small online interaction steps.
>
> $~$
>
> **Q3**: Regarding the off(300k) baseline, does it mean that the initial Offline data, plus the 10k data from online interaction, are used for the final 300k offline gradient steps?
>
> **A3:** Yes, this statement is right. All the off(300k) results, including DOOF and baselines, are obtained using the offline dataset with another 10k online sampled data. During training, half data is sampled from offline dataset while the other half is sampled from the 10k online data.

---

> > ### Comment · Reviewer_pumy · 2024-11-25
> >
> > Thanks for the author’s response. After reading all the reviews and the rebuttal, I will stick with my original score.

---

> > > ### Author Response · Authors · 2024-11-25
> > >
> > > Once again, we express our sincere gratitude for the affirmation of our work and the valuable feedback you have provided!

---

### Official Review · Reviewer_wjRc · 2024-11-08

**Soundness:** 2
**Presentation:** 4
**Contribution:** 2
**Rating:** 6
**Confidence:** 3

**Summary:**

This paper proposes a two-stage approach to address the offline-to-online problem: first, fine-tuning the dynamic model using online data, and then extracting the policy with the MBPO method. Extensive experiments demonstrate that fine-tuning the dynamic model with online data improves the accuracy of estimating true environment dynamics, thereby enhancing policy extraction using MBPO.

**Strengths:**

1) Fine-tuning dynamic models with online data to support model-based RL is an intuitive approach.
2) The paper is well-structured.
3) The experiments effectively illustrate the distribution shift problem and demonstrate the impact of the proposed method.

**Weaknesses:**

1) The proposed method appears to be a technical fine-tuning extension of MBPO, a model-based offline RL method. While the proposed fine-tuning concept could potentially apply to other model-based offline RL methods, this paper focuses solely on MBPO without broader discussion.
2) In Table 1, the authors list key differences between the proposed DOOF and other methods. However, there is the conceptual difference between the baseline methods **did not** attempt offline finetuning and **cannot** be applied to it. The authors should consider the key differences between DOOF and those methods.
3) The experimental results lack convincing evidence. For example, only one seed is used in Table 4, despite Fig. 6(a) showing significant variance during offline fine-tuning. Additionally, there seems to be a statistical mismatch between Table 4 and Fig. 6. Please see detailed feedback in Questions.

**Questions:**

1) What is the value of the coefficient parameter $\lambda_{on}$ in Table 4?
2) There appears to be a statistical mismatch between Table 4 and Figure 6. For example, DOOF achieves 92.5 for HC-m in Table 4 after off(300k) finetuning, but in Figure 6, the highest results fluctuate around 80. Additionally, the pretrained offline result for WK-m in Table 4 is 77.7, yet almost all DOOF results with various coefficient factors are higher than 80. Therefore, using only one seed does not convincingly demonstrate the robustness of the improvement.
3) I appreciate that the authors demonstrate and discuss the dynamics estimation error with different quality datasets under Table 3. Similarly, the final improvements with varying dataset qualities could be discussed in depth. While data are presented in Table 4, it might be more straightforward to compare the average improvement across all tasks (HC, HP, WK) under different dataset qualities (r, m, mr) in a figure. Additionally, the authors mention “medium-expert” in the caption of Table 4 but do not include the results. It would be more comprehensive to include these results in future revisions if such datasets are available.

---

> ### Author Response · Authors · 2024-11-24
> **Response to Reviewer wjRc (1/2)**
>
> Thank you for the valuable comments. We provide clarification on your concerns as follows. We hope these new theoretical and experimental results can address your concerns, and we look forward to your further discussion.
>
> $~$
>
> **W1:** The proposed method appears to be a technical fine-tuning extension of MBPO, a model-based offline RL method. While the proposed fine-tuning concept could potentially apply to other model-based offline RL methods, this paper focuses solely on MBPO without broader discussion.
>
> **A1:** Our proposed decoupled framework is general and indeed applicable to other uncertainty-based offline model-based algorithms. In addition to MOPO, our framework is compatible with Count-MORL [1] and MOBILE [2]. The choice of MOPO is motivated by its simplicity and representativeness.
>
> We have expanded the discussion on the universality of the decoupled framework in Section 3.2 of the new paper.
>
> [1] Kim, Byeongchan, and Min-hwan Oh. "Model-based offline reinforcement learning with count-based conservatism." International Conference on Machine Learning. PMLR, 2023.
>
> [2] Sun, Yihao, et al. "Model-Bellman inconsistency for model-based offline reinforcement learning." International Conference on Machine Learning. PMLR, 2023.
>
> $~$
>
> **W2:** In Table 1, the authors list key differences between the proposed DOOF and other methods. However, there is the conceptual difference between the baseline methods **did not** attempt offline finetuning and **cannot** be applied to it. The authors should consider the key differences between DOOF and those methods.
>
> **A2:** In Table 1, we list the distinctions between our implementation of DOOF and other algorithms. However, the **key differences are highlighted in Table 2**. Existing algorithms for offline-to-online finetuning, including model-based methods, attempt policy improvement while eliminating distributional shift during the online phase. In contrast, within the model-based framework, we have linked the dynamics model finetuning to the elimination of distributional shift. This has led to the decoupled framework that addresses distributional shift through the dynamics finetuning, followed by offline policy improvement.
>
> Furthermore, we fully concur with the reviewer's perspective on Table 1. In order to draw greater attention to the key differences presented in Table 2, we have decided to remove Table 1.

---

> ### Author Response · Authors · 2024-11-24
> **Response to Reviewer wjRc (2/2)**
>
> **W3:** The experimental results lack convincing evidence. For example, only one seed is used in Table 4, despite Fig. 6(a) showing significant variance during offline fine-tuning. Additionally, there seems to be a statistical mismatch between Table 4 and Fig. 6. Please see detailed feedback in Questions.
>
> **Q2:** There appears to be a statistical mismatch between Table 4 and Figure 6. For example, DOOF achieves 92.5 for HC-m in Table 4 after off(300k) finetuning, but in Figure 6, the highest results fluctuate around 80. Additionally, the pretrained offline result for WK-m in Table 4 is 77.7, yet almost all DOOF results with various coefficient factors are higher than 80. Therefore, using only one seed does not convincingly demonstrate the robustness of the improvement.
>
> **A3:** **W3** and **Q2** essentially are the same issue, hence we address them collectively here.
>
> Table 4 (now is the Table 3 in new paper version) actually represents the average across three seeds; due to space limitations, the standard deviations have been omitted.
>
> In Figure 6, we indeed only present the training curve for one seed, resulting in relatively larger fluctuations. But actually, we don't have mismatch. As finetuning is to achieve the best performance as we can,so the results in Table 4 is the average of best evaluation scores of multiple policies that we saved during offline finetuning stage. It's true that some intermediate finetuning policy did indeed achieve better evaluation results at certain moments due to learning better behavior, which is consistent with the best results presented in Table 4.
>
> As for WK-m in Figure 6 (b), the finetuning results with various coefficient factors are higher than 90, not only higher than 80.
>
> $~$
>
> **Q1:** What is the value of the coefficient parameter λon in Table 4?
>
> **A4**: $\lambda_{\text{on}}$ is a hyperparameter that is closely related to the penalty coefficient $\lambda_{\text{off}}$ of offline training. We tune the value of $\lambda_{\text{on}}$ based on the offline penalty coefficient $\lambda_{\text{off}}$, with  $\lambda_{\text{on}}:\lambda_{\text{off}} = 0.25,0.5,1,2$. We have listed $\lambda_{\text{on}}$ values in Table 6 of the appendix.
>
> $~$
>
> **Q3:** I appreciate that the authors demonstrate and discuss the dynamics estimation error with different quality datasets under Table 3. Similarly, the final improvements with varying dataset qualities could be discussed in depth. While data are presented in Table 4, it might be more straightforward to compare the average improvement across all tasks (HC, HP, WK) under different dataset qualities (r, m, mr) in a figure. Additionally, the authors mention “medium-expert” in the caption of Table 4 but do not include the results. It would be more comprehensive to include these results in future revisions if such datasets are available.
>
> **A5:** Thank you very much for your suggestions. we have computed the average improvement across all tasks under different dataset qualities as listed in Table 7 in appendix E. As we can see, our method DOOF outperform all other baselines on Gym tasks.
>
> As for the task on medium-expert, it has achieved near optimal scores during offline stage, leaving limited room for policy improvement. Therefore, these expert datasets are not suitable for offline to online finetuning.

---

> > ### Comment · Reviewer_wjRc · 2024-11-25
> > **Reply to authors**
> >
> > Thank you for the authors' response. The additional results and explanations addressed some concerns, but others, particularly regarding the fairness of comparisons and the robustness of DOOF, remain unresolved. I am open to further discussion but will likely maintain my original score unless these issues are addressed.
> >
> > **4. Table 1 vs. Experiment Implementation (A2 follow up)**
> >
> > The claim in Table 1 seems to not align with the actual experimental setup. In Table 3, the authors report offline finetuning results (i.e., OFF-300K) for other baselines, but Table 1 does not include an offline finetuning stage for these baselines. If the authors also applied Distribution Shift + Policy Improvement in the offline finetuning, Table 1 should be updated to reflect this. I suggest either moving Table 1 to the "Experiments" section for implementation details or replacing it with a textual explanation. Additionally, why is CQL considered to apply distribution shift when it is a model-free method?
> >
> > **5. Fair Comparison and Confidence in Table 3 Results (A3 follow up)**
> >
> > - **a)** Does the implementation technique of "average of best evaluation scores" used in Table 3 also apply to other methods?
> > - **b)** The absence of variance results in Table 3 and the fluctuations in Figure 6 raise concerns about the reliability of DOOF's offline finetuning improvements. Including the variance for DOOF in Table 3 and Figure 6 would help address these concerns. Figure 6(a) suggests that DOOF only yields comparable finetuning results when $\lambda_{on}$ is set to 2. This raises doubts about the robustness of DOOF, as it seems sensitive to hyperparameters. Re-plotting Figure 6 and adding variance data could help clarify this.
> > - **c)** In Q2, I noted that the **pretrained** result for **WK-m** in Table 4 (original manuscript) is 77.7, which appears to correspond to Figure 6(b) at **t=0**. However, the authors stated that finetuned results exceed 90, which is confusing, as Figure 6(b) shows no significant increase and indicates results above 80 (>77.7) at t=0. Could the authors clarify this discrepancy?
> >
> > **6. A4 follow-up**
> >
> > In response to my request for the value of $\lambda_{on}$ in Table 4 (in original manuscript), the provided answer seems not relevant.

---

> > > ### Author Response · Authors · 2024-11-25
> > > **Further Reponse (2/2)**
> > >
> > > > **[ A4 follow-up ]** In response to my request for the value of 𝜆𝑜𝑛 in Table 4 (in original manuscript), the provided answer seems not relevant.
> > >
> > > **A3:** We sincerely apologize for the irrelevant information provided before. The value of $\lambda_{\text{on}}$ has been corrected in Table 5 of the revised paper. Specifically, the values of $\lambda_{\text{on}}$ and the ratio ($\lambda_{\text{on}}:\lambda_{\text{off}}$) across different datasets are as follows:
> > >
> > > | Datasets | $\lambda_{\text{on}}$ | Ratio ($\lambda_{\text{on}}:\lambda_{\text{off}}$) |
> > > | -------- | --------------------- | -------------------------------------------------- |
> > > | HC-r     | 0.5                   | 2                                                  |
> > > | HC-m     | 1.0                   | 2                                                  |
> > > | HC-mr    | 1.0                   | 2                                                  |
> > > | HP-r     | 10.0                  | 2                                                  |
> > > | HP-m     | 2.5                   | 0.5                                                |
> > > | HP-mr    | 5.0                   | 2                                                  |
> > > | WK-r     | 1.5                   | 2                                                  |
> > > | WK-m     | 0.125                 | 0.25                                               |
> > > | WK-mr    | 1.25                  | 0.5                                                |
> > > | P-c      | 1.25                  | 0.5                                                |
> > > | P-h      | 0.25                  | 0.5                                                |
> > >
> > > $~$
> > >
> > > We are grateful for your valuable feedback once again and hope your concern can be addressed!

---

> ### Author Response · Authors · 2024-11-25
> **Further Reponse (1/2)**
>
> Thank you very much for engaging in further discussion with us! With the help of your suggestions and advice, we have once again enhanced the quality of our paper (denoted as **V2**). Below is our clarification about the futher concerns:
>
> $~$
>
> > **[ Table 1 vs. Experiment Implementation (A2 follow up) ]**  The claim in Table 1 seems to not align with the actual experimental setup. In Table 3, the authors report offline finetuning results (i.e., OFF-300K) for other baselines, but Table 1 does not include an offline finetuning stage for these baselines. If the authors also applied Distribution Shift + Policy Improvement in the offline finetuning, Table 1 should be updated to reflect this. I suggest either moving Table 1 to the "Experiments" section for implementation details or replacing it with a textual explanation. Additionally, why is CQL considered to apply distribution shift when it is a model-free method?
>
> **A1:** Strictly speaking, the claim in Table 1 corresponds to the **ON-10K** setting. This is because all the baselines include only the online finetuning without an offline finetuning stage. The **OFF-300K** results for baselines are introduced to make the comparison with DOOF more fair. DOOF finetunes the dynamics model through 10k online interactions and then further improve the policy with an additional 300k offline steps. Comparing DOOF directly with ON-10K would be unfair due to the extra 300k gradient steps.
>
> Furthermore, the purpose of Table 1 is to highlight the innovative aspect of DOOF's framework in 'decoupled' by comparing it with other offline to online algorithms.
>
> CQL is a classical and strong model-free offline to online baseline, although it is not designed for offline to online finetuning. All the offline to online algorithms face the challenge of distributional shift.
>
> $~$
>
> > **[ Fair Comparison and Confidence in Table 3 Results (A3 follow up) ]**
> >
> > **a)** Does the implementation technique of "average of best evaluation scores" used in Table 3 also apply to other methods?
> >
> > **b)** The absence of variance results in Table 3 and the fluctuations in Figure 6 raise concerns about the reliability of DOOF's offline finetuning improvements. Including the variance for DOOF in Table 3 and Figure 6 would help address these concerns. Figure 6(a) suggests that DOOF only yields comparable finetuning results when 𝜆𝑜𝑛 is set to 2. This raises doubts about the robustness of DOOF, as it seems sensitive to hyperparameters. Re-plotting Figure 6 and adding variance data could help clarify this.
> >
> > **c)** In Q2, I noted that the **pretrained** result for **WK-m** in Table 4 (original manuscript) is 77.7, which appears to correspond to Figure 6(b) at **t=0**. However, the authors stated that finetuned results exceed 90, which is confusing, as Figure 6(b) shows no significant increase and indicates results above 80 (>77.7) at t=0. Could the authors clarify this discrepancy?
>
> **A2-a):** Yes, all the finetuning results are the average of the best results with 3 seeds.
>
> **A2-b):** The variance of DOOF has been added in Table 6 of Appendix C.4 and the following table also contains this results. Besides, the Figure 6(a) and (b) have been updated with variance in new paper version.
>
> For dataset HC-m (Figure 6(a)), $\lambda_{\text{on}}$ affect the performance heavily. While for WK-m (Figure 6(b)), the performance is relatively robust with different $\lambda_{\text{on}}$.
>
> | Datasets | DOOF-OFF    | DOOF-OFF-300K |
> | -------- | ----------- | ------------- |
> | HC-r     | 37.52±2.78  | 54.89±3.09    |
> | HC-m     | 70.50±6.12  | 92.48±5.58    |
> | HC-mr    | 69.2±1.62   | 88.28±3.17    |
> | HP-r     | 31.9±0.66   | 32.39±0.64    |
> | HP-m     | 68.28±24.46 | 108.40±1.05   |
> | HP-mr    | 82.61±30.76 | 107.34±0.81   |
> | WK-r     | 4.05±0.34   | 16.85±6.75    |
> | WK-m     | 77.67±15.18 | 93.77±1.27    |
> | WK-mr    | 69.6±11.60  | 97.83±0.80    |
> | P-c      | 48.9±11.89  | 54.19±6.78    |
> | P-h      | 38.66±24.21 | 61.99±11.67   |
>
> **A2-c):** For offline policy training, each epoch contains 1k steps training and one evaluation. So the result at $t=0$ is not the offline pretrained performance. Actually, the $t=0$ is finetuned result with 1k steps, larger than 77.7.
>
> We have recognized the potential for misunderstanding with our previous presentation, and therefore, we have introduced a dashed gray line in Figures 6(a) and 6(b) to represent the offline pretrained results.

---

> > ### Comment · Reviewer_wjRc · 2024-11-26
> > **Reply to authors**
> >
> > I appreciate the additional results provided by the authors, which address my concerns about fair comparison and DOOF's confidence. However, several issues remain unresolved, including the limited improvement from decomposition of DOOF (ON 10k -> OFF 300k, as shown in Figures 6(a) and 6(b)), the significant impact of hyperparameters like $\lambda_{on}$, the contribution of this work (specifically, the discrepancy between what other methods can do and how they are implemented), the "selection of best evaluation results" (in A2-b, and the variance of DOOF-OFF-300k is significantly lower than that of DOOF, despite both being trained on offline datasets). Given the time constraints, I believe these issues cannot be fully addressed, so I will raise my score to 6 and leave it to the other reviewers and PC to determine their significance. Thanks again to the authors for their efforts.

---

> > > ### Author Response · Authors · 2024-11-26
> > >
> > > We are immensely grateful that you have raised your score to 'accept'! Regarding the unresolved issues, below is our discussion on those topics:
> > >
> > > $~$
> > >
> > > > the limited improvement from decomposition of DOOF (ON 10k -> OFF 300k, as shown in Figures 6(a) and 6(b))
> > >
> > > **A1:** In the following table, we have calculated the improvement percentage of DOOF from offline pretrained results to online finetuned results. Except for HP-r, which shows a modest increase of +1.5%, the improvements across other datasets are quite significant. The HC-m dataset in Figure 6(a) exhibits a 31.2% enhancement, while the WK-m dataset in Figure 6(b) demonstrates a 20.7% improvement. The fluctuations during the training process may make these improvements appear less pronounced.
> > >
> > > | Datasets | Improvement of DOOF                 |
> > > | -------- | ----------------------------------- |
> > > | HC-r     | 37.52 $\rightarrow$ 54.89 (+46.3%)  |
> > > | HC-m     | 70.50 $\rightarrow$ 92.48 (+31.2%)  |
> > > | HC-mr    | 69.20 $\rightarrow$ 88.28 (+27.6%)  |
> > > | HP-r     | 31.90 $\rightarrow$ 32.39 (+1.5%)   |
> > > | HP-m     | 68.28 $\rightarrow$ 108.40 (+58.8%) |
> > > | HP-mr    | 82.61 $\rightarrow$ 107.34 (+29.9%) |
> > > | WK-r     | 4.05  $\rightarrow$ 16.85 (+416.0%) |
> > > | WK-m     | 77.67 $\rightarrow$ 93.77 (+20.7%)  |
> > > | WK-mr    | 69.60 $\rightarrow$ 97.83 (+40.6%)  |
> > > | P-c      | 48.90 $\rightarrow$ 54.19 (+10.8%)  |
> > > | P-h      | 38.66 $\rightarrow$ 61.99 (+60.3%)  |
> > >
> > > $~$
> > >
> > > > the significant impact of hyperparameters like 𝜆𝑜𝑛,
> > >
> > > **A2:** $\lambda_{\text{on}}$ plays a crucial role in determining the distribution of data collected through online interactions, which in turn affects the finetuning of both the dynamics model and the policy. Indeed, the selection of $\lambda_{\text{on}}$ is particularly critical in certain environments, such as HC-m (Figure 6(a)) and HP-m (Table 2).
> > >
> > > $~$
> > >
> > > > the contribution of this work (specifically, the discrepancy between what other methods can do and how they are implemented)
> > >
> > > **A3:** Our contribution lies in a decoupled framework that is facilitated through the use of a dynamics model. Consequently, this framework is not applicable to model-free methods. For other model-based approaches, such as FOWM and MOORe, they also do not fully leverage the dynamics model to achieve decoupling.
> > >
> > > $~$
> > >
> > > > the "selection of best evaluation results" (in A2-b, and the variance of DOOF-OFF-300k is significantly lower than that of DOOF, despite both being trained on offline datasets)
> > >
> > > **A4:** The offline pretrained results represent the **final** results of the training. This is because offline training does not permit interaction with the environment, which is necessary for selecting the best results during the training process. However, during the offline to online finetuning phase, the online environment is accessible, making the selection of the **best** evaluation scores acceptable. We hypothesize that the **best** results are more stable compared to the **last** results, leading to a smaller variance.
> > >
> > > $~$
> > >
> > > Lastly, we would like to express our sincere appreciation once again for the recognition of our work. Thank you for the time and effort you have invested in reviewing our paper!

---

### Author Response · Authors · 2024-11-24
**Modification about the paper**

We sincerely appreciate the valuable suggestions and comments provided by the reviewers. In the new version of our paper, we have made several revisions (highlighted in **orange**) to address your concerns and enhance the quality of this paper. Below is the summary of modifications we have made:

**V1**

1. In Section 3.1, we have added Theorem 3.2. Combined with the previous Theorem 3.1, we have completed the theoretical guarantees for online dynamics finetuning;
2. In Section 3.1, we have added Theorem 3.3. With the help of this theorem, we have provided guarantees that offline policy finetuning will lead to performance improvements compared with offline pretrained policy;
3. In Section 3.2, we have added the discussion on the universality of our decoupled framework DOOF;
4. In Figure 3, we have included the missing experimental results, including HP-m, HP-mr, P-c, and P-h;
5. In Table 2, the dataset used to verify online dynamics finetuning has been changed from a fake buffer to a more representative and reasonable dataset uniformly sampled from the state-action space;
6. The results of offline policy fine-tuning in Table 3 have been described more clearly;
7. Appendix A has included additional discussions on the details of Theorems 3.1, 3.2, and 3.3;
8. Appendix D.1 has presented the distribution of the total variation distance across more datasets, serving as a supplement to Figure 1 in the main text;
9. Appendix D.2 has included the distribution of the total variation distance and $u(s,a)$ across more datasets, serving as a supplement to Figure 2 in the main text;
10. Appendix E has added the average improvement of DOOF across different dataset qualities in Table 7.

**V2**

11. In Figure 6 (a) and (b), we have plotted the variance of performance curves. Besides, one dashed gray line is added to represent the offline pretrained result;
12. In Table 5, we have corrected the error in the last column "online exploratory coefficient $\lambda_{\text{on}}$";
13. We have added the Table 6 that contains the std of DOOF.

$~$

We hope that the additional theoretical and experimental results address your concerns and questions. We are always ready and eager to engage in further discussions with you!

---

### Meta-Review · Area_Chair_1CRc · 2024-12-21

**Metareview:**

This paper proposes a decoupled offline-to-online fine-tuning framework (DOOF) for model-based reinforcement learning. It addresses the challenge of distribution shift during the transition from offline to online learning by separating the fine-tuning of the dynamics model from policy optimization. In the online stage, the dynamics model is fine-tuned to mitigate the shift, while policy improvement is conducted offline using the updated model.

The strengths of this paper include a clear intuition for decoupling distribution shift and policy improvement, clear and well-structured presentation, and high performance. The main weaknesses of this paper lie in unresolved issues in the theoretical and experimental parts.

There were intense debates and discussions during and after the rebuttal phase, with one clear rejection and the other borderline positive reviews in the end. The decision is to reject, with details listed in the additional comments below.

**Additional Comments On Reviewer Discussion:**

Before the rebuttal, the initial scores were (3, 3, 5, 5, 6). Finally, the scores are (3, 6, 6, 6, 6).
Given that Reviewers **pumy** and **Moig** did not engage in the post-rebuttal AC-reviewer discussions and provided only brief reviews, their scores (two 6s) were down-weighted when considering the final decision.


During the rebuttal phase:

- **Reviewer wjRc** increased their score from 5 to 6 based on additional results and explanations but still had several concerns.
- **Reviewer pumy** raised questions about the experimental design and maintained their original score.
- **Reviewer Moig** questioned the novelty and increased their score from 5 to 6 during the rebuttal.
- **Reviewer fU5C** raised several questions initially. After multiple rounds of rebuttals and responses, the concerns mainly focused on the correctness of the theoretical aspects and unconvincing experimental settings. However, the reviewer acknowledged the improvements in the revision and the motivation of this work.
- **Reviewer JnJh** increased their score from 3 to 6, as the rebuttal clarified several points regarding the understanding of the theory and framework.

During the post-rebuttal phase, after further discussions from the reviewers, the AC concluded the findings of the reviewers as follows, leading to the rejection of this paper.

- Issues were raised regarding the theory part, especially the newly introduced Theorem 3.2, including:
  - Theorem 21 in [1] (Thm. 21 in the arXiv version, Thm. 18 in the NeurIPS version) is inconsistent with the authors' Equation (13), with a discrepancy between martingale process-generated datasets and a specific state-action pair (Reviewer fU5C).
  - The assumption in Theorem 3.1 regarding the same reward is acceptable (agreed upon by all reviewers), but it is unclear whether DOOF satisfies this assumption (Reviewer fU5C).
  - The authors' omission of a maximum likelihood objective (Equation (6) in [1]) raised concerns (Reviewer JnJh).
  - The symbol in line 3 of Eqn. (14) should be an equals sign (=) instead of $\geq$. Although this typo does not invalidate the equation, it caused confusion.
- Modifying the reward to encourage optimistic exploration (by incorporating an uncertainty reward term) is not new ([2], [3]) (Reviewer fU5C).
- Empirically, the overall improvement from DOOF's core methodological contribution lacks convincing evidence (Reviewer wjRc). For instance, Fig. 6 shows hyperparameters need to be chosen carefully.

Refs:
- [1] Flambe: Structural complexity and representation learning of low rank mdps.
- [2] Ready Policy One: World Building Through Active Learning.
- [3] Optimistic Model Rollouts for Pessimistic Offline Policy Optimization.

---

### Decision · Program_Chairs · 2025-01-22

Reject